# Anthropogenic carbon pathways towards the North Atlantic interior revealed by Argo-O₂, neural networks and back-calculations

Rémy Asselot [1] ✉, Lidia I. Carracedo[1], Virginie Thierry [1], Herlé Mercier [1], Raphaël Bajon [1] & Fiz F. Pérez [2]

The subpolar North Atlantic (SPNA) is a region of high anthropogenic $CO_2$ ($C_{ant}$) storage per unit area. Although the average $C_{ant}$ distribution is well documented in this region, the $C_{ant}$ pathways towards the ocean interior remain largely unresolved. We used observations from three Argo-$O_2$ floats spanning 2013-2018 within the SPNA, combined with existing neural networks and back-calculations, to determine the $C_{ant}$ evolution along the float pathways from a quasi-lagrangian perspective. Our results show that $C_{ant}$ follows a stepwise deepening along its way through the SPNA. The upper subtropical waters have a stratified $C_{ant}$ distribution that homogenizes within the winter mixed layer by Subpolar Mode Water formation in the Iceland Basin. In the Irminger and Labrador Basins, the high-$C_{ant}$ footprint (> 55 µmol kg$^{-1}$) is mixed down to 1400 and 1800 dbar, respectively, by deep winter convection. As a result, the maximum $C_{ant}$ concentration is diluted (<45 µmol kg$^{-1}$). Our study highlights the role of water mass transformation as a first-order mechanism for $C_{ant}$ penetration into the ocean. It also demonstrates the potential of Argo-$O_2$ observations, combined with existing methods, to obtain reliable $C_{ant}$ estimates, opening ways to study the oceanic $C_{ant}$ content at high spatio-temporal resolution.

Since the beginning of the industrial revolution, human activities such as fossil fuel burning and changes in land-use have led to the emission of large amounts of carbon dioxide ($CO_2$) into the atmosphere. This excess of carbon, referred to as anthropogenic carbon ($C_{ant}$), represented an addition of $9.5 \pm 0.5$ GtC yr$^{-1}$ to the atmosphere between 2011 and 2020[1]. The ocean has absorbed $25 \pm 2\%$ of $C_{ant}$ emissions ($170 \pm 35$ GtC out of $660 \pm 65$ GtC of total $C_{ant}$ emissions) since the beginning of the industrial era[1]. The ocean acts, therefore, as a net $C_{ant}$ sink and a moderator of climate change. This net $C_{ant}$ uptake occurs via air-sea exchange, driven by air–sea $CO_2$ disequilibria resulting from the difference between the steady increase in atmospheric $p$CO$_2$ over the years since the pre-industrial era and oceanic $p$CO$_2$[2]. The latter is primarily regulated by seasonal variations of temperature in the subtropics. In contrast, at higher latitudes, its oscillation is typically dominated by biological processes[3] affecting dissolved inorganic carbon (DIC) and total alkalinity ($A_T$). A low DIC/$A_T$ ratio will lead to a large $CO_2$ uptake capability by the ocean in response to an increase in atmospheric $p$CO$_2$[4]. As a result of the accumulation of $C_{ant}$, the Revelle factor, measuring the buffer capacity of the ocean, decreases[5]. Ultimately, changes in oceanic circulation also have the capacity to modify the saturation of $C_{ant}$ at the ocean surface, enhancing or dampening the ocean carbon uptake capability[6].

The global $C_{ant}$ distribution in the ocean is not homogeneous, neither vertically nor horizontally[7]. As expected from direct air-sea exchange, $C_{ant}$ concentration is maximum at the surface and decreases with depth. Of the global ocean, the North Atlantic Ocean (Fig. 1) is the region with the largest $C_{ant}$ inventory per surface area[8,9], storing up to 23–38% of the total oceanic $C_{ant}$[2,10]. Within this region, $C_{ant}$ uptake

[1]University of Brest, Ifremer, CNRS, IRD, Laboratory of Spatial and Physical Oceanography (LOPS), 29280 Plouzané, France. [2]Institute of Marine Investigations (IIM, CSIC), 6 Eduardo Cabello Street, 36208 Vigo, Spain. ✉e-mail: remy.asselot@ifremer.fr

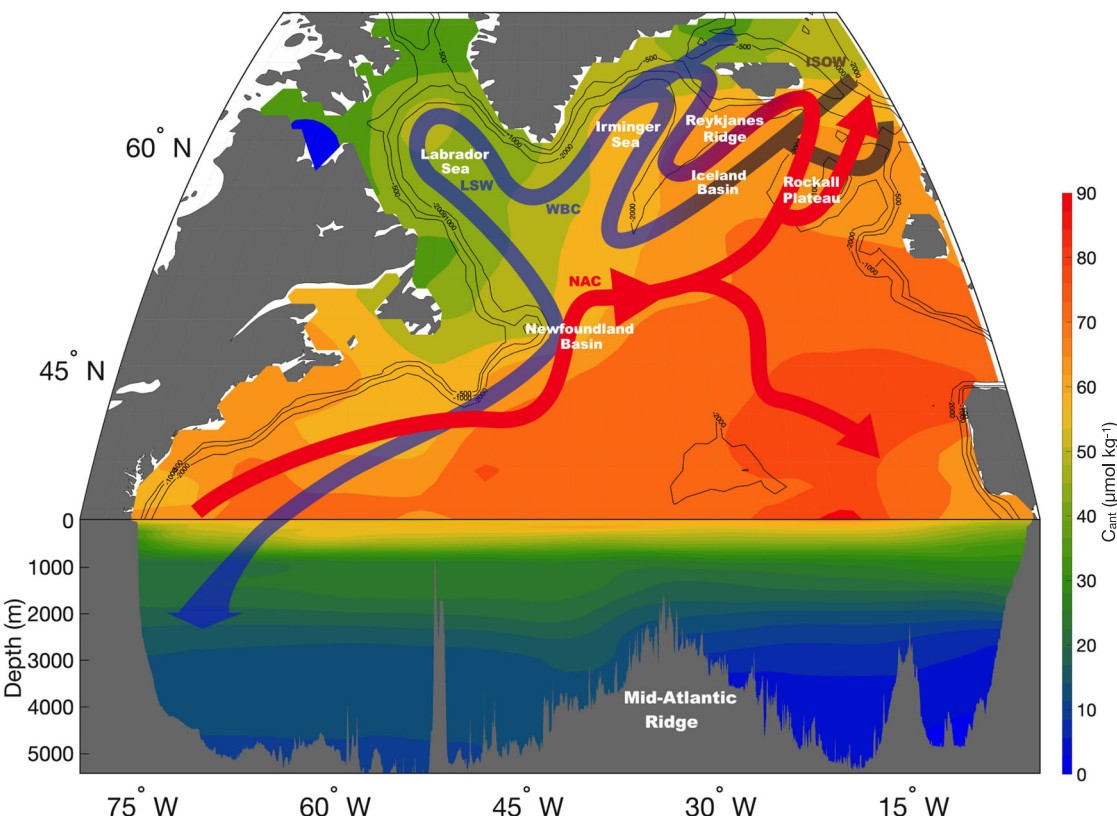

**Fig. 1 | Main circulation patterns and $C_{ant}$ distribution (µmol kg⁻¹) for the year 2015 in the subpolar North Atlantic gyre.** Red arrows represent the upper limb of the Atlantic Meridional Oceanic Circulation (AMOC) while blue arrows indicate the lower limb of the AMOC. The purple arrow represents ocean circulation between the upper and lower limb of the AMOC. The black contours represent the isobath −500 m, −1000 m, and −2000 m. The section on this figure depicts the whole water column, while our study focuses on the top 2000 m of the water column only. NAC North Atlantic Current, WBC Western Boundary Current, ISOW Iceland-Scotland Overflow Water, LSW Labrador Sea Water. $C_{ant}$ concentrations come from the GLODAPv2 dataset[56] and were normalized to the year 2015 using the exponential equation of Carter et al.[25].

mostly occurs in the subtropics[6,7] due to the low Revelle factor of subtropical surface waters[2]. These subtropical $C_{ant}$-loaded waters are then transported northward toward subpolar latitudes[6,11] by the upper branch of the Atlantic Meridional Overturning Circulation (AMOC), where additional $C_{ant}$ uptake takes place[12]. The subpolar North Atlantic (SPNA) gyre has been identified as one of the most important regions for the injection of $C_{ant}$ towards the deep ocean[7]. This deep $C_{ant}$ penetration is mainly due to the formation of Labrador Sea Water (LSW) through deep winter convection in the Labrador and Irminger Seas[7,13–16]. Considering the role of the ocean in moderating the ongoing climate change through its capacity to uptake and store $C_{ant}$, any change in its uptake and storage rates might have drastic consequences on the climate system. Changes in the $C_{ant}$ storage rates in the North Atlantic Ocean are fundamentally related to the increased atmospheric $p\text{CO}_2$, the changes in the AMOC strength, the changes in the North Atlantic Oscillation (NAO) index, and deep water formation[6,16,17]. The current $C_{ant}$ increasing trends and related $C_{ant}$ storage rates might thus be modified in the future due to the increasing atmospheric $C_{ant}$, and the projected weakening of both AMOC[18] and deep-water formation[19,20].

To date, $C_{ant}$ estimates are mainly based on methods that rely on scarce but valuable ship-based measurements of carbonate system parameters (carbon-based methods[2,6,21–23]) or transient tracers such as CFCs (transient tracer-based methods[9,16]). However, studying the spatio-temporal evolution of oceanic $C_{ant}$ storage and understanding the processes involved is a crucial challenge that requires a more detailed view of the upper and deep $C_{ant}$ distribution and of its main pathways into the ocean interior. Additionally, the distribution of $C_{ant}$ on timescales shorter than GO-SHIP cruises is necessary to understand

the effect of $\text{CO}_2$ emissions reduction and carbon removal strategies on the ocean. Considering the unrivaled spatio-temporal sampling provided by the Argo-$O_2$ network[24], the purpose of this study is to demonstrate that Argo-$O_2$ data combined with existing neural networks (i.e., ESPER_NN[25], CANYON-B and its routine CONTENT[26]) and a back-calculation method ($\varphi C_T^O$ method[17,27]) can be used to obtain reliable $C_{ant}$ estimate at the finest spatio-temporal scale to date. As a case study, we selected three Argo-$O_2$ floats to describe $C_{ant}$ deepening as it enters and propagates along the SPNA region. We highlight the long journey of $C_{ant}$ from the surface towards the interior of the SPNA gyre. Our results open up perspectives to investigate $C_{ant}$ storage changes and transport at seasonal to decadal timescales with the growing global and seasonally unbiased Argo-$O_2$ data network compared to ship-based measurements.

## Results

### Distribution of $C_{ant}$ associated with the North Atlantic Current

Four main regions with the different vertical distributions of $C_{ant}$ were identified along the floats' pathways: region 1, located East of the Grand Banks of Newfoundland, and regions 2, 3, and 4, comprising the Iceland, Irminger, and Labrador basins, respectively (Fig. 2a). Region 1 is located inside a dynamic transition zone separating the subpolar and subtropical domains of the North Atlantic Ocean[28]. This transition zone is typically characterized by a sharp thermohaline and density front, which, in this case (Fig. 2c), indicates that float 5904988 entered the North Atlantic Current (NAC)[29]. The thermohaline front is also accompanied by a sharp $C_{ant}$ and oxygen gradient (Fig. 2b and Supplementary Fig. 5). In the first 400–600 dbar of the water column, region 1 is characterized by saline, warm, and poorly oxygenated North

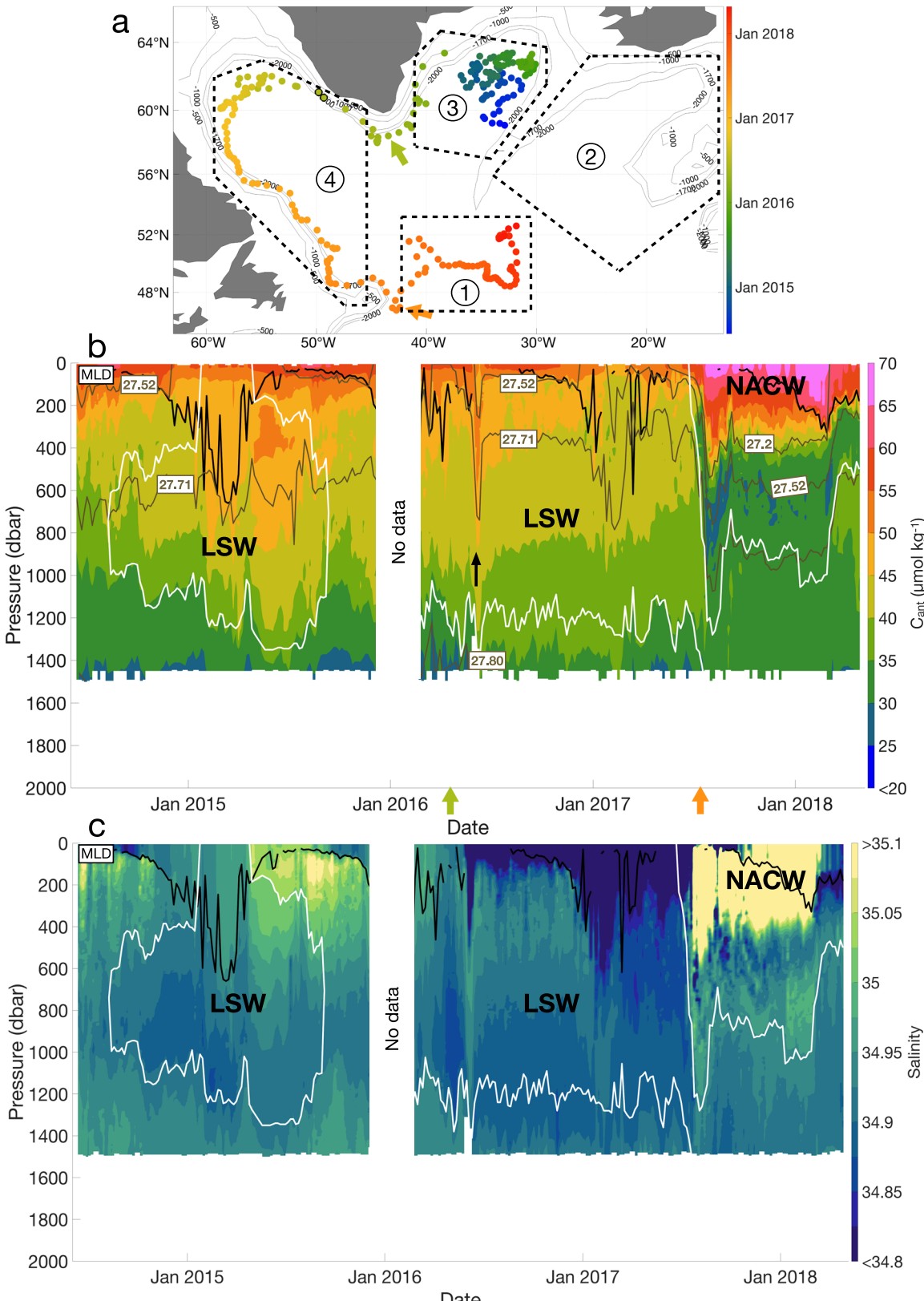

**Fig. 2 | Characteristics of the Argo float 5904988 in the North Atlantic (47-65°N; 15-65°W). a** Trajectory of the float with the four main regions defined. The color coding represents the date, with blue being the beginning of the trajectory and red the end. The green arrow indicates the moment the float passes Cape Farewell, and the orange arrow indicates when the float enters Region 1. The points with black outlines correspond to the Argo profiles identified as belonging to a passing eddy (see main text). **b** Section of estimated anthropogenic carbon ($\mu mol\ kg^{-1}$) along the float pathway. The black line represents the mixed layer depth (MLD). The white lines represent the limits of the Labrador Sea Water (LSW), defined by $O_2 \geq 290\ \mu mol\ kg^{-1}$ in the Labrador and Irminger Seas and by $S < 34.94$ outside these two basins. The brown lines represent the isopycnals ($kg\ m^{-3}$). The black arrow indicates when the float is located within an eddy. LSW Labrador Sea Water, NACW North Atlantic Central Water. **c** Salinity section along the float pathway.

Atlantic Central Water (NACW) of subtropical origin. At this location, the NACW comprises the northern extension of the upper and thermocline subtropical waters, defined as water masses located between the surface and $\sigma_\theta = 27.0$ kg m$^{-3}$ [12]. These waters are typically characterized by a low Revelle factor[5], which favors atmospheric $C_{ant}$ uptake along their journey from the subtropics. As a consequence, the NACW presents the highest $C_{ant}$ values (maximum $C_{ant} > 65.0$ µmol kg$^{-1}$) of all water masses identified along the pathway of float 5904988 (Fig. 2a). The values of $C_{ant,def}$ close to zero (Supplementary Fig. 3) indicate that the NACW is saturated in $C_{ant}$ due to the air-sea $C_{ant}$ uptake occurring during the journey of the water masses that form the NACW. Furthermore, just eastward of the outcrop of the isopycnal 27.2 kg m$^{-3}$, the high $C_{ant}$ concentrations ($C_{ant} > 55.0$ µmol kg$^{-1}$) reach the base of the winter mixed layer depth (MLD) at ~400 dbar (Fig. 2b). In this region, the high $C_{ant}$ signal is vertically transferred as the MLD deepens, highlighting the vertical $C_{ant}$ homogenization due to mixing during winter convection. Below the mixed layer, between August 2017 and March 2018, $C_{ant}$ concentration in region 1 decreased sharply to a relative minimum of $26.2 \pm 6.6$ µmol kg$^{-1}$ near 800 dbar (Fig. 2b). These concentrations increase again up to $36.1 \pm 7.5$ µmol kg$^{-1}$ at 1050 dbar, associated with the Labrador Sea Water (LSW). As inferred by the progression of the float, the $C_{ant}$-loaded NACW is transported northwards by the NAC (Fig. 1), toward the Iceland Basin (region 2), following the general circulation pattern of the SPNA gyre[30].

## Distribution of $C_{ant}$ in the Iceland Basin and over the Reykjanes Ridge

Region 2 encompasses the Iceland Basin, including the eastern flank of the Reykjanes Ridge, and is characterized by the presence of Subpolar Mode Water (SPMW) in the first 600 dbar of the water column. In region 2, the SPMW contains the largest $C_{ant}$ values over the water column, whose average is $57.2 \pm 5.8$ µmol kg$^{-1}$. The SPMW is formed during winter convection[31,32] by intense air–sea buoyancy loss, leading to the densification of the $C_{ant}$-loaded NACW on its way toward subpolar latitudes[33]. This process is well illustrated by the good agreement between the winter MLD and the depth of the SPMW base (Figs. 3b and 4b). In the Iceland Basin, the depth of the isoC$_{ant}$ 50 µmol kg$^{-1}$, which mostly follows the winter MLD, reaches up to 600 dbar (Figs. 3b and 4b), that is, 200 dbar deeper than in region 1. Below the SPMW, $C_{ant}$ concentration decreases rapidly, highlighting trapping of $C_{ant}$ within the SPMW and revealing that the formation of this water mass is a precursor in the deepening of $C_{ant}$ in the subpolar gyre. To further assess this deepening, Fig. 5 illustrates the mean $C_{ant}$ concentration along four density ranges describing the main water masses found in the four regions[12,34]. In the second-density layer ($27.2 < \sigma_\theta < 27.52$ kg m$^{-3}$) of region 2, the averaged $C_{ant}$ concentration along the float 6901023 pathway increases from $40.1 \pm 6.2$ µmol kg$^{-1}$ in the south-easternmost part of the region to $55.5 \pm 6.5$ µmol kg$^{-1}$ on the east side of the Reykjanes Ridge (Fig. 4b). This east-west $C_{ant}$ enrichment is likely due to the trajectory of the float, going from an area containing a thin SPMW layer, to an area with a thick SPMW layer (Fig. 3b). In this region, $C_{ant,def}$ fluctuates between $-20$ µmol kg$^{-1}$ and 0 µmol kg$^{-1}$ (Supplementary Fig. 3), suggesting that air–sea $C_{ant}$ enrichment takes place, explaining part of the east–west $C_{ant}$ enrichment in the Iceland basin. In the western part of Region 2, the Reykjanes Ridge entails a natural topographic barrier separating the Iceland and Irminger basins, two basins with different $C_{ant}$ distribution and water masses. The isoC$_{ant}$ 35 µmol kg$^{-1}$, which fluctuated between 600 and 800 dbar east of the Reykjanes Ridge, deepens until ~1200 dbar on the western side of the ridge (Figs. 3b and 4b). The second density layer of region 2 ($27.2 < \sigma_\theta < 27.52$ kg m$^{-3}$) disappears west of the ridge during the winter months (Fig. 5, Supplementary Figs. 6c and 7c) due to outcropping and subsequent surface water densification in the vicinity of the ridge[12]. As a consequence, the third oceanic layer

($27.52 < \sigma_\theta < 27.71$ kg m$^{-3}$) shallows on the western side of the Reykjanes Ridge, illustrated by the shallowing of the isopycnal 27.71 kg m$^{-3}$ from ~1000 dbar east of the ridge to less than 500 dbar west of the ridge (Fig. 5b, c). Along with this densification of the surface water, $C_{ant}$ increases by 34% and 31% in the third layer for float 6901023 and float 6901026, respectively, marking a rapid transfer of $C_{ant}$ into the third oceanic layer. However, if it cannot be excluded that the $C_{ant}$ increase is only due to vertical entrainment, this increase could partly be due to lateral entrainment of water masses from another origin. These results highlight the role of the Reykjanes Ridge, separating the $C_{ant}$ distribution and dynamics between region 2 and region 3[34]. In the fourth oceanic layer ($27.71 < \sigma_\theta < 27.80$ kg m$^{-3}$), $C_{ant}$ concentrations are the lowest and slightly increase along the float trajectories (Fig. 5b, c). This increase stays within the averaged uncertainty range ($\pm 5.9$ µmol kg$^{-1}$) and is therefore not significant.

## Deep transport of $C_{ant}$ in the Irminger and Labrador basins

The Irminger Sea (region 3) is characterized by high concentrations of $O_2$ (Supplementary Figs. 5b, 6b, and 7b), delimiting the core of the LSW[34]. These large $O_2$ values are indicative of the intense air-sea exchange that characterizes this region[13,35,36]. In contrast to regions 1 and 2, here, the vertical $C_{ant}$ distribution has been subject to a large water-column homogenization. This leads to a weaker $C_{ant}$ gradient, with a gradual decrease of mean $C_{ant}$ concentrations from $52.0 \pm 5.8$ µmol kg$^{-1}$ at the surface to $41.2 \pm 7.9$ µmol kg$^{-1}$ at the lower bound of the LSW (~1400 dbar) (Figs. 2b and 4b). The high $C_{ant}$ values are constrained within the LSW, meaning that the formation of this water mass traps and transports $C_{ant}$ toward the deeper ocean. Yet, the maximum $C_{ant}$ concentrations are still found at the surface, with values reaching $52.0 \pm 5.8$ µmol kg$^{-1}$ (Figs. 2b and 4b). As suggested by the negative $C_{ant,def}$ in the surface waters of the Irminger Sea (Supplementary Fig. 3b and 3c), these high surface concentrations are likely due to the intense air–sea $CO_2$ exchange, following the same behavior of $O_2$ exchanges[35]. Following the Western Boundary Current (WBC) (Fig. 1), $C_{ant}$ is transported out of the Irminger Sea. In the WBC, we observe the occurrence of occasional high-$C_{ant}$ pulses throughout the water column (black arrows on Figs. 2b and 3b) that we relate to anticyclonic mesoscale eddies (Supplementary Fig. 8). These anticyclonic features lead to a punctual downward isopycnal displacement of surface waters containing high $C_{ant}$ concentration, explaining the $C_{ant}$ pulses identified in our sections. Such eddies could carry $C_{ant}$ away from their formation regions and when they collapse, they might contribute to the isopycnal mixing of this tracer. Once in the Labrador Sea, float 6901023 illustrates how the isoC$_{ant}$ 35 µmol kg$^{-1}$ reaches a maximum pressure level of 1800 dbar in the inner part of the basin (Fig. 3b). That is, implying a ~ 200 dbar deepening of this isoC$_{ant}$ compared to what is observed in the Irminger Sea. As in the Irminger Sea, in the Labrador Sea, the high $C_{ant}$ concentrations are also contained within the LSW, confirming that the formation of this water mass is a first-order deep-trapping mechanism for $C_{ant}$.

## Discussion

This study describes the $C_{ant}$ evolution along its pathways from the upper to the intermediate layers (up to 2000 dbar) of the SPNA gyre. It is based on a combination of existing neural networks and the back-calculation $\varphi C_T^O$ method, using Argo-$O_2$ observations as input data. Our results are based on three floats following the main branches of the subpolar gyre circulation. From this quasi-lagrangian perspective, we identified four main $C_{ant}$ deepening stages and related them to their corresponding physical mechanisms. At the southernmost boundary of our study region, in the subtropical/subpolar transition zone near Flemish Cap, $C_{ant}$ is mainly trapped within the NACW layer ($\sigma_\theta < 27.2$ kg m$^{-3}$). The NACW has the largest mean $C_{ant}$ concentration ($62.8 \pm 5.9$ µmol kg$^{-1}$) of all water masses intercepted by the studied floats. At this location, winter mixing favors vertical $C_{ant}$

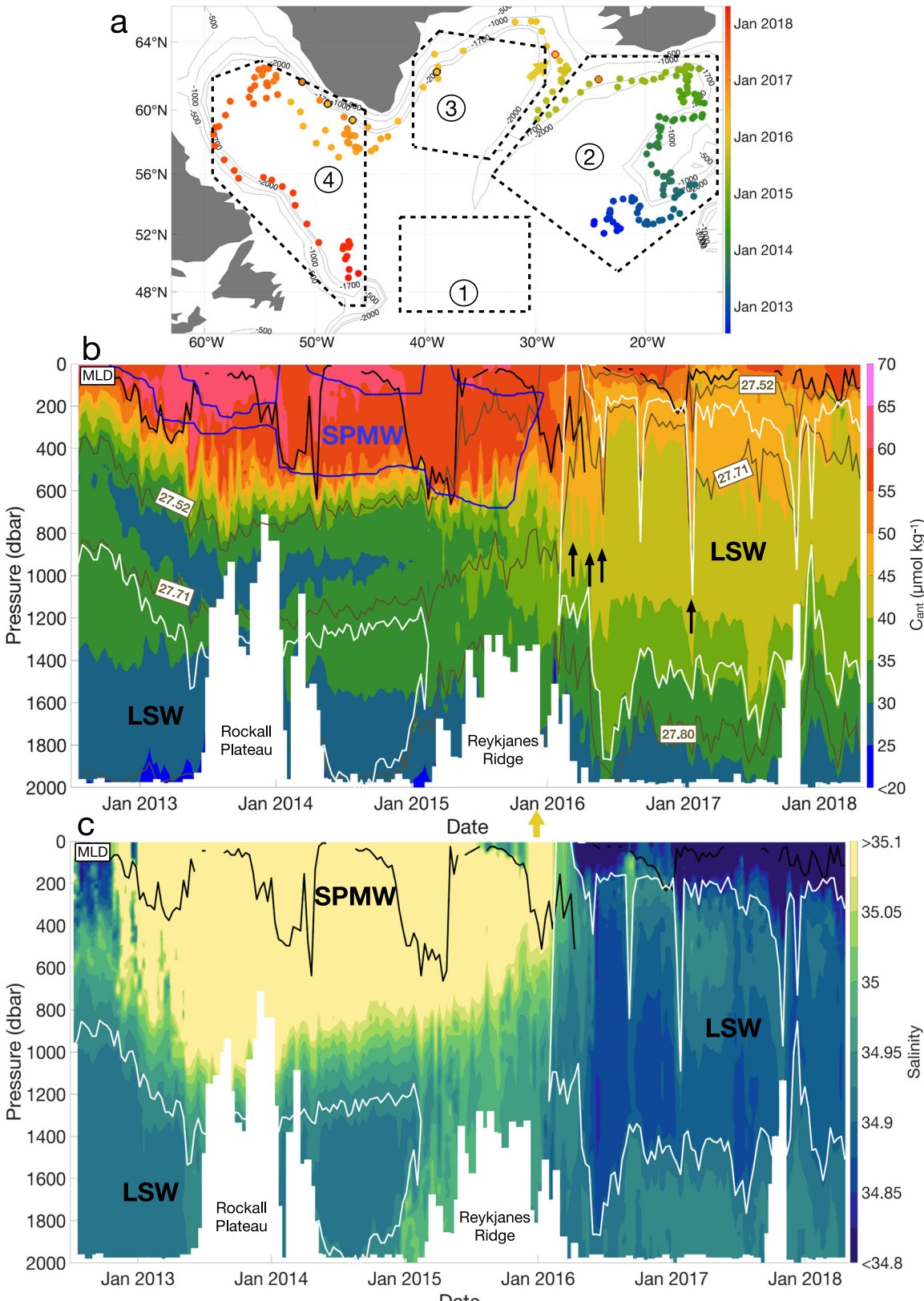

**Fig. 3 | Characteristics of the Argo float 6901023 in the North Atlantic (47–65°N; 15–65°W). a** Trajectory of the float with the four main regions. The color coding represents the date, with blue being the beginning of the trajectory and red being the end. The yellow arrow indicates when the float has passed the Reykjanes Ridge. The points with black outlines correspond to the profiles pointed out by the black arrows on the $C_{ant}$ section. The points with red outlines represent the limits of the Reykjanes Ridge, defined by a depth shallower than 1700 m. **b** Section of

$C_{ant}$ (µmol kg⁻¹) along the float pathway. The black line represents the MLD. The white lines represent the limits of the Labrador Sea Water (LSW), defined by $O_2 \geq 290$ µmol kg⁻¹ in the Labrador and Irminger Seas and $S < 34.94$ outside these two basins. The blue lines represent the limits of the Subpolar Mode Water (SPMW) defined by a potential vorticity lower than $6 \times 10^{-11}$ m⁻¹ s⁻¹. The brown lines represent the isopycnals (kg m⁻³). Black arrows indicate when the float is located in an eddy. **c** Salinity section along the float pathway.

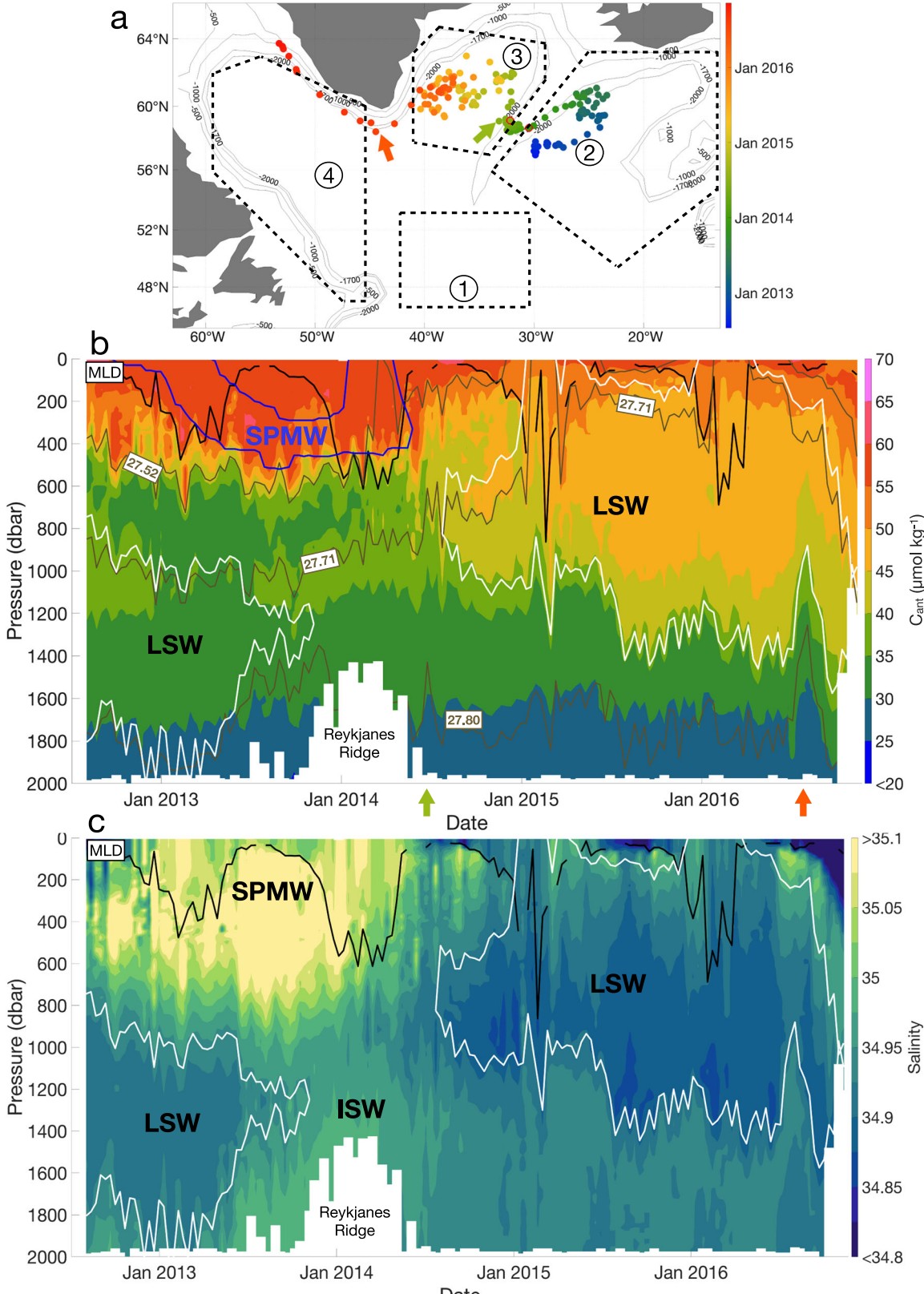

**Fig. 4 | Characteristics of the Argo float 6901026 in the North Atlantic (47–65°N; 15–65°W). a** Trajectory of the float with the four main regions. The color coding represents the date, with blue being the beginning of the trajectory and red being the end. The green arrow indicates when the float has passed the Reykjanes Ridge, while the red arrow shows when the float passes Cape Farewell. The points with red outlines represent the limits of the Reykjanes Ridge, defined by a depth shallower than 1700 m. **b** Section of $C_{ant}$ (µmol kg⁻¹) along the float pathway. The black line represents the MLD. The white lines represent the limits of the Labrador Sea Water (LSW), defined by $O_2 \geq 290$ µmol kg⁻¹ in the Labrador and Irminger Seas and $S < 34.94$ outside these two basins. The blue lines represent the limits of the Subpolar Mode Water (SPMW) defined by a potential vorticity lower than $6 \times 10^{-11}$ m⁻¹ s⁻¹. The brown lines represent the isopycnals (kg m⁻³). **c** Salinity section along the float pathway.

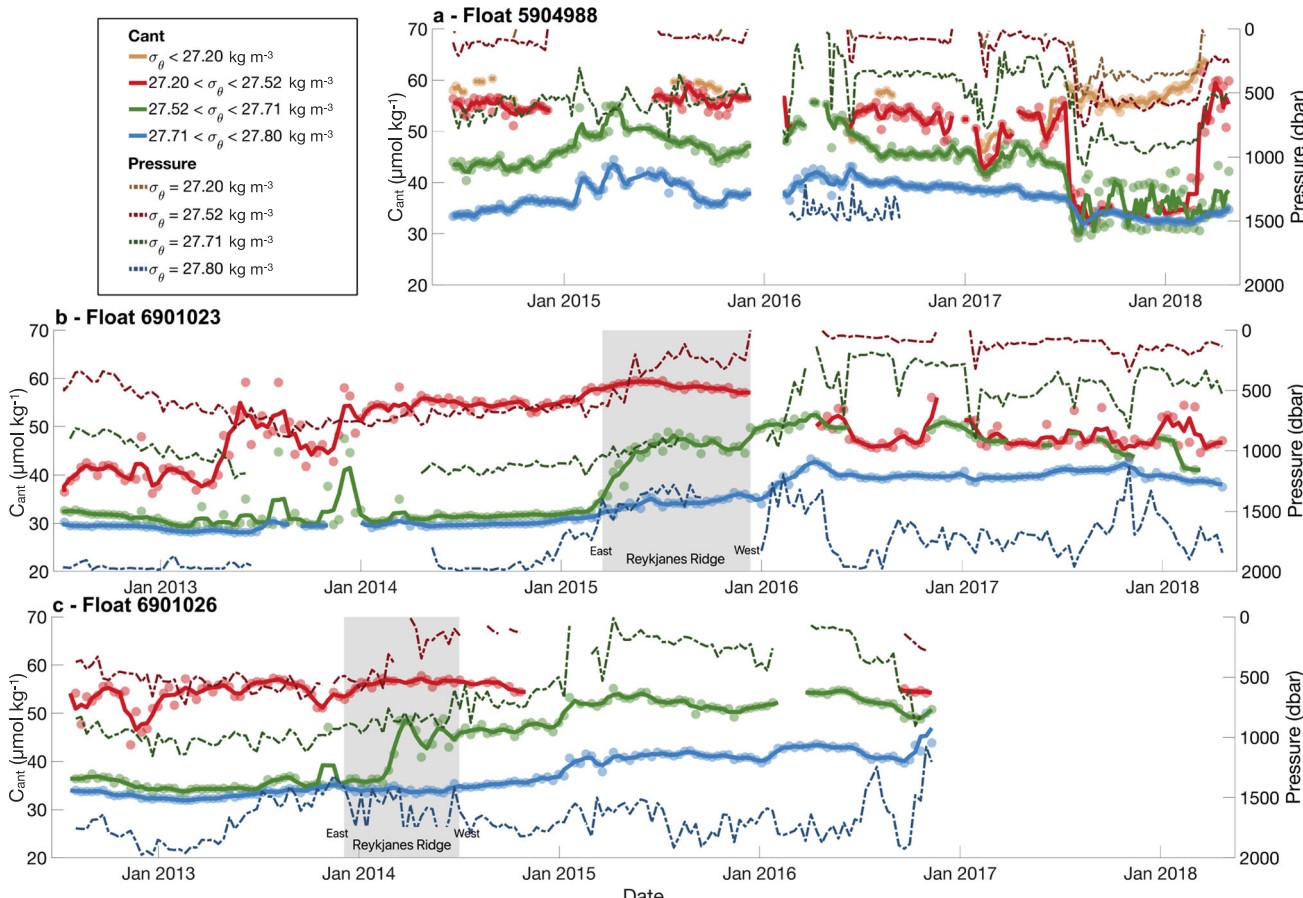

**Fig. 5 | Averaged $C_{ant}$ concentrations (µmol kg$^{-1}$) over density layers. a** For float 5904988. **b** For float 6901023. **c** For float 6901026. Scatter points indicate raw $C_{ant}$ data, while the solid lines represent the monthly moving mean of $C_{ant}$ concentrations. Dashed lines represent the pressure levels of the isopycnals. The gray rectangle illustrates the time period when the floats are above the Reykjanes Ridge.

homogenization, by which high $C_{ant}$ concentrations (>45.0 µmol kg$^{-1}$) reach down to ~400 dbar (1$^{st}$ deepening stage). The air–sea heat fluxes and mixing processes along the northward progression of the $C_{ant}$-loaded NACW lead to a densification of NACW and its transformation into SPMW as it reaches the Iceland Basin. This transformation causes a deepening of $C_{ant}$ down to ~600 dbar in this region (2$^{nd}$ deepening stage). The average $C_{ant}$ concentration within this newly formed SPMW (57.2 ± 5.8 µmol kg$^{-1}$ for $\sigma_\theta$ < 27.52 kg m$^{-3}$) is lower than upstream due to the $C_{ant}$ content being spread out in the vertical. Further west, the Reykjanes Ridge is identified as a key topographic feature that marks a steep transition between the Iceland Basin and the Irminger Basin with markedly different vertical $C_{ant}$ distributions. In the Irminger Sea, the $C_{ant}$ signal (>35.0 µmol kg$^{-1}$) deepens until 1400 dbar (3$^{rd}$ deepening stage) and is constrained within the LSW layer, highlighting the role of the LSW formation on the $C_{ant}$ deepening. The deepening is even more pronounced in the Labrador Sea, where the $C_{ant}$ signal, still confined within the LSW, reaches 1800 dbar (4$^{th}$ deepening stage). The newly formed LSW joins the Deep Western Boundary Current as part of the North Atlantic Deep Water (NADW), spreading the $C_{ant}$ footprint downstream towards the ocean interior. Further evidence of this $C_{ant}$ trapping within the LSW and its propagation downstream is provided by our data east of the Flemish Cap transition zone, where a relative $C_{ant}$ maximum can be identified within the LSW layer underneath the NACW (Fig. 2b). As surface waters spread through the different basins of the SPNA, we also inferred water masses that have favorable conditions to uptake additional atmospheric $C_{ant}$, by means of the $C_{ant,def}$ proxy. Our results indicate that air-sea $C_{ant}$ fluxes are likely to occur all over the region, and as far west as in the Labrador Basin.

Deep water formation is a broadly known and acknowledged mechanism for $C_{ant}$ penetration in the interior of the North Atlantic Ocean[2,6,7]. Relying on Argo-O$_2$ data and existing methods, we presented an original approach that allowed us to describe this $C_{ant}$ deepening at the finest spatiotemporal scale to date. Our results revealed the role of the SPMW formation as an early stage of subpolar $C_{ant}$ deepening. This result reinforces the idea that $C_{ant}$ deepening occurs throughout the whole SPNA gyre and is not only linked to the main convective sites (i.e., Irminger and Labrador Basins), as traditionally attributed[7,16]. Mainly uptaken from the atmosphere in the subtropics[2], $C_{ant}$ is transported northwards towards the SPNA region by the NAC[6], which at these latitudes is the main component of the upper AMOC limb. The quasi-lagrangian Argo-float trajectories are a suitable and timely means for describing this horizontal $C_{ant}$ journey. Despite lateral advection by the upper AMOC limb, being the main source of $C_{ant}$ in the region, the contribution from ocean $C_{ant}$ uptake is also relevant[6,12]. By means of the $C_{ant,def}$ proxy, the Argo-O$_2$ data also support the hypothesis of surface air-sea $C_{ant}$ enrichment all along the subpolar gyre domain by an outcropping of the low-carbon nutrient stream isopycnal range[12]. Given the dominant role of lateral advection in $C_{ant}$ distribution, it is not surprising that AMOC variability affects the northward $C_{ant}$ transport towards the SPNA and its storage rates[6,11]. Under current climate change, the AMOC is projected to weaken[37], and we could thus expect a concomitant decrease in $C_{ant}$ storage rates and content in the SPNA. Nevertheless, the northward oceanic $C_{ant}$ transport might still be subject to continuous increase in response to the rise in anthropogenic CO$_2$ emissions[11,38]. Estimates on a finer spatiotemporal scale, such as those based on Argo-O$_2$ observations, should

be sustained in the long term to determine which effect will be dominant in the future.

Our case study has shown that neural networks combined with high-quality in situ Argo-$O_2$ measurements and a back-calculation method can effectively be used to retrieve $C_{ant}$ concentration through the three-dimensional estimates of oceanic variables (nutrients, DIC, total alkalinity). These oceanic variables are needed to compute $C_{ant}$ at a higher spatio-temporal resolution than that available from the current biogeochemical (BGC) observation networks. Hence, the methodology presented here, enchaining Argo-$O_2$ observations and existing methods, opens up a promising step forward to study $C_{ant}$ distribution, its variability, and the processes driving these variations, which are crucial challenges regarding the ocean carbon sink[39]. The global nature of the Argo-$O_2$ array gives the opportunity to significantly increase the spatio-temporal coverage of current $C_{ant}$ estimates. However, we can only guarantee the validity of our approach in the case-study region and for the period (1970–2020) over which the ship-based data used for training and validation of the neural networks were collected (GLODAPv2.2020[40]). Neural networks can only reproduce what they have learned. Therefore they must be trained with new data as the ocean changes, which is particularly true for the SPNA gyre where interannual to decadal changes are considerable (e.g., cold blob[41] or great salinity anomaly[42]). The importance of high-accuracy in situ data measurements for training the neural networks cannot be disregarded, as such, the continuation of ship-based observations should be assured. Another important consideration of our results and their interpretation lies within the averaged total uncertainty of our method, which is ±5.9 μmol kg$^{-1}$. Any spatio-temporal change in $C_{ant}$ concentration below two times this value should be interpreted with caution. This current methodological uncertainty could possibly be reduced by using data from the growing BGC-Argo database[43,44] to train the neural networks. Moreover, BGC-Argo floats, equipped with pH and nitrate sensors, could be directly used to derive $C_{ant}$, hence providing independent validation $C_{ant}$ estimates when both BGC-Argo and Argo-$O_2$ floats are concomitant in space–time. However, BGC-Argo floats were not included in our study because available floats did not match our selection criteria. In view of the above, there is, therefore, a complementarity between the scarce high-quality ship-based BGC measurements and the high spatiotemporal BGC-Argo and Argo-$O_2$ observations. This complementarity is even more marked because Argo data need to be adjusted (QCed) against ship-based data. The Argo-$O_2$ float data presented in this case study reached a maximum pressure level of 2000 dbar, limiting our result to the top 2000 dbar of the water column. The OneArgo mission calls for the deployment of 1250 floats capable of reaching 4000 or 6000 dbar[45]. Using Deep-Argo floats equipped with dissolved oxygen sensors could thus strengthen our findings and improve our knowledge about the mechanisms controlling the deep/abyssal $C_{ant}$ transport. For instance, the role of the Iceland-Scotland Overflow Water (ISOW), flowing below 2000 dbar, could not be identified here even though this water mass is known to contain $C_{ant}$[11].

Finally, it is worth mentioning that Earth system models tend to misrepresent the vertical penetration of $C_{ant}$ due to the shallow and weak formation of NADW[46], even if this issue is better addressed in the latest generation of CMIP6 models[18,47,48]. The results of this study provide insights subject to improving the representation of the key pathways for the deep $C_{ant}$ propagation in Earth system models. Implementing a better representation of these pathways would definitely ameliorate the simulated vertical distribution of $C_{ant}$ and, ultimately, improve climate predictions.

## Methods
### Argo float observations
To highlight the $C_{ant}$ penetration into the deep ocean, we use Argo-$O_2$ floats, giving the opportunity to obtain one $C_{ant}$ profile every 10 days. We selected the Argo-$O_2$ floats that (1) followed the cyclonic pathway

of the SPNA gyre, (2) had a lifetime longer than 3 years, and (3) crossed the A25 OVIDE hydrographic section[49], which will serve as a reference dataset for validation. Three floats matched these criteria, comprising a total of 651 profiles of pressure (P), temperature (T), salinity (S), and oxygen ($O_2$)[50] within the region 47–65°N; 15–65°W (Supplementary Fig. 1). Data span from July 2012 to April 2018. Only data adjusted in delayed mode with a quality flag of 1 or 2 (good or probably good data)[51] were used in our analysis (Supplementary Table 1). The accuracy of the data is assumed to be 0.002 °C, 0.01, and 2.4 dbar for temperature, salinity, and pressure, respectively[52], and better than 3 μmol kg$^{-1}$ for oxygen[53].

### Estimating biogeochemical variables with neural networks
To derive the biogeochemical variables (nutrients, DIC, and $A_T$) needed to compute $C_{ant}$, we rely on predictive neural networks. Neural networks (NN) are machine learning algorithms based on a multi-layer perceptron[54,55], that are trained and validated against observations. The input data of the neural networks are, in this study, Argo P/T/S/$O_2$ measurements. We used two neural networks, namely, ESPER_NN[25] and CANYON-B, associated with its routine CONTENT[26]. ESPER_NN was used to obtain the macronutrients (phosphate, nitrate, and silicate) using P, T, S, $O_2$, location, and time as predictors. ESPER_NN was adopted over other ESPER methods (i.e., ESPER_LIR and ESPER_Mixed) because it gives the lowest biases and root mean square errors over the global ocean for the predicted macronutrients[25]. ESPER_NN reproduces the validation dataset with average biases and errors of ~2% for macronutrients (Supplementary Table 1)[25]. However, in the North Atlantic Ocean, ESPER_NN gives uncertainties of ~1.3% for the predicted carbonate variables ($A_T$ and DIC), which is higher than the previous NN. Consequently, $A_T$ and DIC were computed with CANYON-B[26], and the outputs of CANYON-B were passed through the CONTENT routine. This routine ensures consistency between carbonate variables and thus reduces the uncertainties of the carbonate system variables to ~0.5% for $A_T$ and DIC (Supplementary Table 1). We used neural networks rather than climatological products (such as, e.g., GLODAPv2 climatology[56]) because gridded climatologies are unable to capture changes in carbon variables in regions where water masses move laterally due to mesoscale processes or rapid circulation changes. This is because climatologies represent the mean field in which the variability has been smoothed out. In contrast, neural networks, such as ESPER_NN and CANYON-B, derive biogeochemical variables based on water mass characteristics, hence they can cope with such changes. In particular, the use of Argo-$O_2$ float data as input to the neural networks reinforces this water-mass change tracking capability due to the quasi-lagrangian behavior of the Argo-$O_2$ floats and their temporal resolution, three times greater than monthly climatological products.

### Anthropogenic carbon estimates
To determine the anthropogenic carbon fraction ($C_{ant}$) from DIC, we used the carbon-based back-calculation $\varphi C_T^O$ method[17,27]. This method has been widely applied to study the inventory of $C_{ant}$, its storage rates, its variability[14,17,57], and the influence of $C_{ant}$ on ocean acidification[58]. In any back-calculation method, $C_{ant}$ is computed as a two-step approach: (1) the changes in DIC due to biological activity occurring since a water parcel has left the ocean surface are removed, and (2) the pre-industrial preformed DIC is also removed, with the residual being interpreted as $C_{ant}$. The $\varphi C_T^O$ method (Supplementary Equation 1) presents two main advantages compared to other back-calculation methods. First, this method accounts for the spatiotemporal variability of the preformed $A_T$. Second, the parameterization of the preformed $A_T$ and disequilibrium terms are determined using the subsurface layer as a reference[27]. These two modifications improve the $C_{ant}$ estimates in cold and deep water formation regions subject to strong mixing processes, such as the SPNA gyre. A study comparing observational methods to estimate $C_{ant}$ in the Atlantic Ocean, including the TTD[9], the

TrOCA[59], the C°$_{IPSL}$[60], the $\Delta C^*$[61], and the $\varphi C_T^O$ method[17], demonstrated that the latter provided the closest value to the average of all methods for the whole latitudinal range[27]. Based on that, we selected the $\varphi C_T^O$ method to perform our analysis. The input variables for this method are date, geographical location, T, S, O$_2$ (in this study, the Argo variables), the macronutrients (obtained from ESPER_NN), plus A$_T$ and DIC (computed from CANYON-B and CONTENT). The $\varphi C_T^O$ method requires date as input because it adjusts the oceanic C$_{ant}$ concentration against the corresponding atmospheric $pCO_2$. It also requires the geographical location (latitude and longitude) to define the water masses involved in the computation of the preformed A$_T$ and disequilibrium terms. All the input variables have associated uncertainties that propagate through C$_{ant}$ calculations. The uncertainties on Argo pressure, temperature, and salinity data are 2.4 dbar, 0.002 °C, and 0.01, respectively[52]. These uncertainties relate to the instrument's performance. The uncertainties on O$_2$ data are about 3.5 µmol kg$^{-1}$ (Supplementary Table 1). The uncertainties of the predicted variables A$_T$, DIC, and macronutrients are, on average, 10.85, 10.48, and 0.73 µmol kg$^{-1}$, respectively (Supplementary Table 1). These NNs uncertainties are provided as part of the NNs predictive outputs and they result from the combination of training measurement uncertainties, the Bayesian method uncertainties, and the NNs input sensitivities[26]. Taking all of them into account, we quantified the uncertainty associated with our C$_{ant}$ estimate by randomly generating 100 C$_{ant}$ fields (Supplementary Table 1) using a Monte Carlo method[62]. The standard deviation from these C$_{ant}$ fields fluctuates between ±5.4 µmol kg$^{-1}$ and ±10.2 µmol kg$^{-1}$ (Supplementary Fig. 2) and the overall average value (for the three floats) is ±5.9 µmol kg$^{-1}$. It is worth highlighting that the uncertainties of the modeled tracer fields (Supplementary Table 1) are evidently greater than the corresponding in situ measured variables (i.e., by 1.5 µmol kg$^{-1}$ for oxygen). However, the uncertainties of the predicted variables are yet reasonably small to serve as inputs to the back-calculation method. Consequently, they allow an estimation of C$_{ant}$ that is statistically significant (and whose final uncertainties lie within 9% to 19% of the signal). To further validate our approach, we compared these ArgoTSO$_2$-NN-based C$_{ant}$ estimates with ship-based C$_{ant}$ estimates obtained during OVIDE cruises[49,63]. We identified Argo-O$_2$ profiles within the 35 km radius and less than 7 month-timespan of an OVIDE CTD cast (Supplementary Fig. 2). For each CTD cruise cast, C$_{ant}$ is estimated from in situ bottle-sample measurements of nutrients, oxygen, A$_T$ and pH, via the $\varphi C_T^O$ method (standard procedure[11]). Emulating the approach proposed in this study for the Argo-O$_2$ floats (TSO$_2$-NN procedure), we re-estimated C$_{ant}$ by using the ship-based P/T/S/O$_2$ data as input to the neural networks to obtain nutrients, DIC, and A$_T$, and then applied the $\varphi C_T^O$ method. These C$_{ant}$ profiles (ship-based standard and ship-based TSO$_2$-NN) were then compared to our Argo-based C$_{ant}$ estimates (Argo-based TSO$_2$-NN procedure), and all of them agreed well within the averaged method uncertainty of ±5.9 µmol kg$^{-1}$ (Supplementary Fig. 2). This comparison indicates that the C$_{ant}$ values obtained from ship-based measurements and C$_{ant}$ estimates calculated via Argo-O$_2$ data and neural networks give consistent results.

We also computed a proxy that allows the indirect assessment of the air−sea C$_{ant}$ uptake capacity, the C$_{ant}$ deficit (C$_{ant,def}$, Supplementary Fig. 3). This proxy was defined by Ridge & McKinley[12] as the difference between the C$_{ant}$ concentration of a water parcel and a reference ($\langle C_{ant} \rangle_{\sigma_\theta < 26.5}$) computed as the average C$_{ant}$ concentration between the surface and the potential density ($\sigma_\theta$) of 26.5 kg m$^{-3}$. This isopycnal delimits the subtropical surface waters assumed to be in equilibrium with the current atmosphere. A water parcel with a negative C$_{ant,def}$ value indicates that this parcel has a deficit of C$_{ant}$. If transported to the surface, this water parcel will have a lower $pCO_2$ compared to a water parcel without this deficit, hence favoring the air−sea C$_{ant}$ uptake. Usually, C$_{ant,def}$ becomes more and more negative with depth (Supplementary Fig. 3), meaning that the deep waters

would uptake a higher amount of C$_{ant}$ compared to the subsurface waters if they were transported to the ocean surface. The reference value is computed for 2002 with the GLODAPv2 dataset[56], considering the averaged C$_{ant}$ concentration in surface waters ($\sigma_\theta < 26.5$ kg m$^{-3}$) offshore the Venezuelan coasts (10–20°N; 70–64°W), and is $\langle C_{ant} \rangle^{2002}_{\sigma_\theta < 26.5}$ = 51.5 µmol kg$^{-1}$. We assume that our reference water mass offshore the Venezuelan coasts takes 5 years to arrive in the Iceland and Irminger basins[64]. For a given float, we use that reference value rescaled in time to 5 years before the time of its first profile, using the exponential equation of Carter et al.[25]. Assuming that due to its lagrangian behavior, the float follows the transformation of the same water mass, we kept the same reference value along the float trajectory within the subpolar gyre. However, this assumption is not valid for float 5904988 when it drifted from Region 4 to Region 1 and entered the North Atlantic Current, where we re-initialized the reference value.

## Mixed layer depth and water mass definition

To determine the mixed layer depth (MLD) along the Argo float trajectories, we use the threshold method, which is based on a density difference of 0.01 kg m$^{-3}$[65] between the surface and the base of the mixed layer[66]. Since this method tends to overestimate the MLD during deep convection events in winter[67], we conducted a visual inspection[68] to verify the MLD values during these events. When the MLD values disagreed between the threshold method and the visual inspection, we kept the value determined by visual inspection.

The North Atlantic is composed of several water masses (Fig. 1), most of which can be identified by their potential temperature ($\theta$), S, $\sigma_\theta$, O$_2$, and potential vorticity. Here, the potential vorticity is calculated via the Brunt−Väisälä frequency, the gravity, and the Coriolis parameter. In the upper layers, the North Atlantic Current (NAC) flowing from the subtropics to the subpolar regions (Fig. 1), carries North Atlantic Central Water (NACW)[30]. The lower limit of this water mass is defined by $\sigma_\theta = 27.2$ kg m$^{-3}$[69] (Supplementary Fig. 4). Continuous air-sea interaction and winter convection along the NAC path cools and freshens the NACW, leading to its transformation into Subpolar Mode Waters (SPMWs)[31,32]. SPMWs are characterized by nearly homogeneous properties, with S > 34.98, $\sigma_\theta < 27.71$ kg/m$^3$, and a potential vorticity lower than $6 \times 10^{-11}$ m$^{-1}$ s$^{-1}$[31,70] (Figs. 3 and 4). At intermediate depths (400−2000 dbar), in the subpolar gyre, the Labrador Sea Water (LSW) is located between $27.71 < \sigma_\theta < 27.80$ kg/m$^3$. The LSW is formed by deep winter convection in the Labrador Sea[71] and Irminger Sea[36,65], trapping large amounts of O$_2$ and C$_{ant}$ during its formation[72]. In those basins, we defined the LSW by O$_2 \geq 290$ µmol kg$^{-1}$[34]. However, along its path, the LSW loses O$_2$ due to mixing and biological activity. Therefore the O$_2$ criteria is not fully reliable to identify the LSW core away from its source regions. As a consequence, outside the Irminger and Labrador Seas we identify the LSW by S < 34.94[53]. At deeper levels (up to 2000 dbar), the Iceland-Scotland Overflow Water (ISOW) flows along the Reykjanes Ridge and is identified by $\sigma_\theta > 27.80$ kg m$^{-3}$ and S > 34.94[34] (Supplementary Fig. 4).

## Eddy identification

To detect the eddies along the Argo-float pathways, we run the autonomous eddy identification scheme of Faghmous et al.[73], which monitors mesoscale ocean eddy activity in global sea level anomaly (SLA) dataset (http://www.aviso.altimetry.fr/duacs/). The scheme starts from the simple notion that every eddy has a single extremum, defined as a grid cell whose SLA is higher or lower than its 24 neighbors in a $5 \times 5$ grid. To determine the contour of the eddy, the algorithms search for the largest possible contour that would allow the feature not to violate the assumption that an eddy can have only a single extremum within its interior. For each eddy feature identified at time $t$, the eddies at time $t + 1$ are searched to find the closest feature within a predefined search space. When a feature at time t is associated with another feature at time $t + 1$, their sizes are compared to ensure that

they are reasonably similar from a physical point of view. There are two major uncertainties associated with this eddy identification method. First, the use of a geometric eddy definition, as opposed to a physical one, introduces uncertainties in the eddy boundaries because they are not necessarily associated with the physical properties of the eddy. Second, as the method constrains features to have a single extremum, small features that are in close proximity might cancel each other out and not be detected. Despite these limitations, the success discovery rate of the autonomous eddy identification algorithm is 96.4%[73].

### Reporting summary

Further information on research design is available in the Nature Portfolio Reporting Summary linked to this article.

## Data availability

The original Argo data can be freely downloaded on the Euro Argo Data Selection platform (https://dataselection.euro-argo.eu/). Our $C_{ant}$ estimates, needed to evaluate the conclusion of the paper, can be downloaded on Zenodo (https://doi.org/10.5281/zenodo.7071614).

## Code availability

The Matlab codes used for the simulation are available from the corresponding author upon request.

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

## Acknowledgements

R.A. has received funding, as part of the EuroSea project, from the European Union's Horizon 2020 research and innovation program under grant agreement No. 862626. L.I.C., V.T., and R.B. acknowledge support from Ifremer. H.M. was supported by CNRS. F.F.P. was supported by the BOCATS2 (PID2019-104279GB-C21) project funded by MCIN/AEI/ 10.13039/501100011033. This work is a contribution to CSIC's Thematic Interdisciplinary Platform PTI WATER:iOS. The authors gratefully acknowledge financial support by the Brittany Region for the CPER Bretagne ObsOcean 2021-2027 and from the French government within the framework of the "Investissements d'avenir" program integrated in France 2030 and managed by the Agence Nationale de la Recherche (ANR) under grant agreement no ANR-21-ESRE-0019 for the Equipex+ Argo-2030 project. The data were collected and made freely available

by the International Argo Program and the national programs that contribute to it (https://argo.ucsd.edu, https://www.ocean-ops.org). The Argo Program is part of the Global Ocean Observing System. The GLODAPv2 database is freely available as a numeric data package at the Carbon Dioxide Information Analysis Center (CDIAC) http://cdiac.ornl.gov/oceans/GLODAPv2. The Ssalto/Duacs altimeter products were produced and distributed by the Copernicus Marine and Environment Monitoring Service (http://www.marine.copernicus.eu).

## Author contributions

R.A., L.I.C., and V.T. designed and developed the concept of the study. R.A. conducted the data analysis with inputs from L.I.C., V.T., H.M., R.B., and F.F.P. R.A. drafted the first version of the paper. All co-authors read and reviewed the paper, and all co-authors agreed on the final version of the paper.

## Competing interests

The authors declare no competing interests.
