## [Peer Review File · Nature Communications]

REVIEWER COMMENTS

Reviewer #1 [Withdrawn without report]

Reviewer #2 (Remarks to the Author):

Asselot et al. present a unique approach utilizing Argo float data and neural networks to investigate the pathways of anthropogenic carbon from the surface to the deep North Atlantic. The authors have put in much effort in providing a better understanding of the role of water mass transformation and winter convection for Cant transport to the deeper ocean. Although the scientific scope of the study is intriguing and the results are noteworthy, there are several areas where the article could be improved. Therefore, I strongly recommend that major revisions be carried out to improve the overall quality of the article.

I give a general assessment of the study and provide line-by-line comments to help improve the manuscript.

I am surprised that the authors opted to use CONNECT for DIC and AT instead of GLODAPv2, which covers DIC and AT, as well as nutrients and oxygen in the same locations. I have serious reservations about the reliability of NN algorithms in accurately representing the seasonal changes in carbonate system parameters, particularly in areas with mesoscale processes like eddies, currents, and fronts. Therefore, the authors may have added avoidable uncertainty to their analysis. Also, the method of calculation of anthropogenic carbon and the equations used are missing. For example, the study used the nominal year 2015 for their analysis but didn't provide enough detail as to why. Even though the previous literature is cited, the equation should be provided in the supplementary material.

Line 24: Authors claim Cant distribution is well documented but still go ahead to say the deep distribution remains largely unresolved. Please rephrase to avoid confusion for readers.

Line 25: Argo data and NN? Neural network observation was critical to your findings. This should be stated in your abstract.

Line 29: Remove the hyphen in high- Cant. You also need to be more explicit when using the word "high." There is a need to give a reference so that readers have an idea of the Cant concentration or magnitude of the surrounding water with respect to the west of the Reykjanes ridge.

Line 33: I am concerned about the choice of your word "For the first time." The method is reliable, but it isn't new (carbon-based back-calculation method by Pérez et al. (2008) and Vazquez-Rodriguez et al. (2009), and it is not the first time work has been done in the North Atlantic (Davila et al., 2022). Using new combined data observations (Argo float and NN) differs from the method itself (which isn't new).

Line 52: Again, be careful with using verbs like "fast and slow" when used without context or reference. I understand the point you are trying to make, but maybe using a word like "atmospheric pCO₂ has steadily increased over the years since the preindustrial era compared to the oceanic pCO₂. You can go further to quote a figure and provide a reference.

Line 53: I understand what you were trying to write, but it should be clear that seawater pCO₂ is primarily regulated by seasonal variations in temperature (in the subtropics), whereas at higher latitudes, its oscillation is typically dominated by biological processes (e.g., photosynthetic CO₂ fixation and the remineralization of organic carbon) (Takahashi et al., 2009; Ko et al., 2021). DIC and TA are two of the four marine carbonate systems (the other two are pH and pCO₂) which we use to constrain the carbon cycle. A change in these two will definitely change pCO₂.

Takahashi, T., Sutherland, S. C., Wanninkhof, R., Sweeney, C., Feely, R. A., Chipman, D. W., et al. (2009). Climatological mean and decadal change in surface ocean pCO₂, and net sea-air CO₂ flux over the global oceans. *Deep Sea Res. Pt. II* 56, 554–577. doi: 10.1016/j.dsr2.2008.12.009

Ko Y, Park G-H, Kim D and Kim T-W (2021) Variations in Seawater pCO₂ Associated With Vertical Mixing During Tropical Cyclone Season in the Northwestern Subtropical Pacific Ocean. *Front. Mar. Sci.* 8:679314. doi: 10.3389/fmars.2021.679314

Line 55: For clarity, you should link how the high (low) surface AT (DIC) concentration relates to the low seawater pCO₂ and the subsequent uptake of atmospheric pCO₂. Provide references where necessary.

Line 60-61: The global Cant distribution in the ocean is not homogeneous, neither vertically nor horizontally. Reference?

Line 106: Cross-check again. The reference year in both Carter et al. (2021) and Lauvset et al. (2016) is 2002. Where is 2015 from?

Line 114: I recommend you add a figure showing the three floats used and their trajectories represented in different colors.

Line 120: Support with Table S1

Line 135: Provide the biases and errors for the nutrients from the NN referred to here.

Line 136: Instead of referring readers back and forth between papers, you should provide the equations used in the supplementary materials.

Line Line 179-181: Looking at the figures from Line 563 (E.D Fig 4), “the negative Cant,def values (blue color) indicate that this particular parcel has a deficit of Cant and is able to uptake Cant from the atmosphere”. State what the positive values (orange color) represent too. Also, for example, in Fig 4C, there are more negative Cant,def values at depths below compared to around the surface to 800 dbar. Does that mean the deep can still take up as much as -15 μmol/kg? State clearly in the text where this is first mentioned.

Line 193: Reading up to this line, I see the mix up at line 106. Your data spans from July 2012 to April 2018; why did you choose the nominal year 2015?

Line 214: intermediate depth. State the depth range.

Line 223-225: Quote figure to support.

Line 277: Authors need to give a detailed highlight of the role of ocean circulation (not just mention “Daniault et al., 2016”) in the transport of Cant from the surface to the deep waters and the implication for the carbon cycle in the event of climate change (enhanced/reduced circulation).

Line 306 and 307: Consistency with using Fig. and Figs when highlighting more than one figure. (Revise entire manuscript).

Line 311: Despite the data spanning five years (July 2012 to April 2018). One significant interpretation missing from this study is the interannual variability (not even mentioned once) and possible implications for the future of the Cant and carbon cycle.

Line 318: “In this region Cant,def < 5.0 ± 7.6 $\mu\text{mol/kg}$, a value which stays within the method uncertainty (Extended Data Fig. 4)”. Revise the statement. Did you mean Cant,def is below 5.0 ± 7.6 $\mu\text{mol/kg}$?

Line 436: Add “profiling float” to Argo observation.

Line 436-437: Highlight the potential feedback mechanisms between the deep transport of Cant and climate change, and how will this impact the ocean's ability to absorb and store anthropogenic carbon in the future?

Line 443: The methodology utilized data from three different sources Argo float (P, T, S, O₂), ESPER_NN (Nutrients), and CONTENT (DIC and AT), and this contributed to uncertainties that would have been reduced if the data was from a singular insitu source or platform. Highlight the importance of reducing uncertainty for future studies by using a more wholesome platform like the Biogeochemical Argo, which in addition to the P, T, S, and O₂, also has nitrate and pH, from which DIC and AT can be estimated with far less uncertainty. Consider citing the relevant literature.

Claustre, H., Johnson, K.S., Takeshita, Y., 2020. Observing the Global Ocean with Biogeochemical-Argo. *Annu. Rev. Mar. Sci.* 12, 23–48. <https://doi.org/10.1146/annurev-marine-010419-010956>

Addey, C.I. Using Biogeochemical Argo floats to understand ocean carbon and oxygen dynamics. *Nat Rev Earth Environ* 3, 739 (2022). <https://doi.org/10.1038/s43017-022-00341-5>

Lastly, the authors heavily relied on the NN estimates for their analysis, but this isn't captured in the manuscript title and keywords (only mentions Argo-O₂ floats). Authors should revise the manuscript title to reflect the combined Argo float and NN approach.

Reviewer #3 (Remarks to the Author):

Dear Rémy Asselot and colleagues,

It was a pleasure to read and think about your manuscript entitled “Argo-O₂ floats reveal the anthropogenic carbon pathways towards the deep North Atlantic”, in which the ocean interior distribution of anthropogenic carbon (Cant) is derived from oxygen, salinity, temperature and pressure measurement obtained from three Argo floats that travelled through major parts of the North Atlantic Ocean from 2013 through 2018. The reconstructed patterns in Cant are further related to the distribution of water masses and previous knowledge about their transport, such that transport processes of Cant can be inferred.

I consider this study very timely and important, given the current challenge of the ocean carbon community to resolve ocean interior carbon dynamics at high resolution. This challenge appears particularly relevant when considering the rapid changes in the ocean carbon cycle expected under

declining CO₂ emissions and the potential implementation of carbon dioxide removal activities. Your study has the potential to contribute significantly to tackling this challenge. It is mostly transparently described and provides - with few exceptions - sufficient methodological details. The results appear solid as presented, the text is well written and structured, and figures are clear and informative.

However, a few (interconnected) points may require major revisions to ensure the correct interpretation and contextualisation of the results, which is important for the community to derive precise conclusions concerning the future application of this method and inherent limitations. I organised my main concerns into four sections below, while detailed comments are provided in the attached pdf file.

Data selection

This study is based on a selection of three Argo floats providing observations from the North Atlantic between 2013 - 2018. Thus, the results provide a snapshot of the Cant distribution during this period, while temporal dynamics are not resolved in this study. My impression is that this nature of the results needs to be expressed more clearly in the abstract to avoid raising false expectations. Given that the float data as used in this study allows only for a snapshot in time, I was wondering about the added value of using the float data and not a static climatology of the input variables, which would permit resolving spatial patterns over a three-dimensional grid covering the whole study region. Maybe the advantages of using the float data could be described more clearly to motivate this study. In this regard, I deem it also important to inform the reader about the criteria based on which the three floats were selected. Specifically, you may want to address why this study does not use the full fleet of Argo-O₂ profiles from this region in order to gain a more complete picture of the Cant distribution in space. In addition, I was surprised that this study focuses exclusively on O₂ as a biogeochemical input parameter for the determination of the CO₂ system and - based on that - the Cant concentration. Wouldn't float-based pH measurement provide a valuable (additional) constraint on the CO₂ system? I imagine that the availability of float pH data in the study region might not be sufficient to base the results of this study on pH measurements. Nevertheless, it appears important to address the potential use of pH measurements to guide future Argo deployments and research on the topic.

Uncertainty estimates

A bulk uncertainty of the Cant estimates ($\pm 7.6 \mu\text{mol/kg}$) is provided that is based on a Monte Carlo approach obtained from varying the input parameters of the Cant calculation within their ranges of uncertainty. If my understanding of this procedure is correct, then the primary outcome should be an individual uncertainty estimate associated with each calculated Cant value. These individual uncertainty estimates should vary with the uncertainty of the input parameters. This understanding evokes a couple of questions: (1) Is it justified and appropriate to average the individual uncertainties into a single bulk value that is then reported identically for each Cant estimate? My impression is that it would be more informative to provide specific uncertainty estimates for each Cant value, such that one can distinguish results with higher and lower confidence. (2) Does the uncertainty estimate include a specific contribution that arises from CO₂ system calculations? In this regard it would also be informative to know how the $\phi^{\text{C}}\text{T}$ method differs when it is provided with AT and DIC (TSO₂-NN approach) or AT and pH (standard approach) data. Please note that this comment links also to my first general comment regarding the use of directly measured Argo pH data. (3) According to table S1, the uncertainty of DIC as an input parameter is $>10 \mu\text{mol/kg}$ for all three floats. How is it possible that the uncertainty of the derived Cant estimate is lower than that uncertainty of the input parameter?

Applicability of the method to achieve unprecedented temporal resolution of the Cant propagation into the ocean interior

While the methods used in this study allows to provide a snapshot of the Cant distribution in the 2010s and link this distribution to the presence of water masses and their transport, I'm doubtful if it will enable us to substantially increase the temporal resolution with which we can track the propagation of Cant into the ocean interior. My main concern in this regard arises from the uncertainty inherent to the determination of Cant. Assuming that the uncertainty estimate provided in this study ($\pm 7.6 \mu\text{mol/kg}$) is correct, then the uncertainty is of similar magnitude as the decadal changes in the concentration of Cant that the upper ocean experiences at current (still high!) rates of increase in atmospheric CO₂. It is thus questionable whether sub-decadal changes in the storage of Cant can be resolved with the method as presented in this study. Given that decadal-scale changes in the Cant storage can be determined from ship-based measurements as well, I encourage the authors to revise their conclusions that "unprecedented" temporal resolution can be achieved and provide a more thorough assessment of this anticipated skill.

I'm further wondering how convinced the authors are of the skill of their method to separate changes in anthropogenic and natural DIC. Imagining for example that the surface ocean was losing natural DIC and oxygen due to warming, would you expect that two subsequent Cant estimates obtained with the $\phi^{\text{C}}\text{T}$ method would be perturbed by co-occurring changes in these variables or not?

Emphasis on eddy pumping

I was fascinated by the co-location of anticyclonic eddies and the deep extension of high Cant concentrations identified in this study. It is a prime example how the frequent and repeated Argo observations help to better understand the cycling of carbon in the ocean interior. As this topic is currently restricted to a few sentences in the results and discussion, I encourage the authors to put a bit more emphasis on these exciting findings. Some lead questions that might be worth considering when expanding on this topic include: Are all anticyclonic eddies detected along the float trajectories associated with a deep penetration of Cant, or are some eddies not effective? How does the strength of the deep penetration signal relate to the uncertainty of the Cant estimate? Can it be excluded that the deep Cant penetration is an artefact of changes in the O₂ distribution induced by the eddies?

I hope you find this feedback helpful to improve some aspects of your study. Please do not hesitate to get in touch if any comments are not clear to you.

REVIEWER COMMENTS

Reviewer #2 (Remarks to the Author):

Asselot et al. present a unique approach utilizing Argo float data and neural networks to investigate the pathways of anthropogenic carbon from the surface to the deep North Atlantic. The authors have put in much effort in providing a better understanding of the role of water mass transformation and winter convection for C_{ant} transport to the deeper ocean. Although the scientific scope of the study is intriguing and the results are noteworthy, there are several areas where the article could be improved. Therefore, I strongly recommend that major revisions be carried out to improve the overall quality of the article.

We would like to thank the reviewer, we appreciate the positive comments as well as the questions raised. Those comments were helpful to revise and improve our work.

I give a general assessment of the study and provide line-by-line comments to help improve the manuscript.

I am surprised that the authors opted to use CONNECT for DIC and AT instead of GLODAPv2, which covers DIC and AT, as well as nutrients and oxygen in the same locations. I have serious reservations about the reliability of NN algorithms in accurately representing the seasonal changes in carbonate system parameters, particularly in areas with mesoscale processes like eddies, currents, and fronts. Therefore, the authors may have added avoidable uncertainty to their analysis.

In areas where water masses move laterally due to mesoscale processes or circulation changes, gridded climatology such as GLODAPv2 (Lauvset et al., 2016) cannot capture the implied changes in carbon variables because this climatology represents the mean field in which the variability has been smoothed out. However, neural networks, such as CONTENT, which derived carbon variables based on water mass characteristics, can cope with these changes.

To prove these statements, we show here a zoom of an Argo-NN-based C_{ant} section where an eddy is localized (second black arrow on Fig. 3) and the corresponding C_{ant} section generated with GLODAPv2 data (Fig. A below). The Argo section shows a deepening of C_{ant} where the anticyclonic eddy is located while this is not visible on the corresponding C_{ant} section generated with GLODAPv2 data. You can also refer to Fig. S7 showing a better agreement of our Argo-NN-based C_{ant} profiles with ship-based C_{ant} estimates than with GLODAPv2-based C_{ant} profiles.

Fig A: Left panel: Argo-based C_{ant} section showing the effect of an anticyclonic eddy on the C_{ant} distribution. Data is from the float 6901023 and covers the month of March 2016. Right panel: Corresponding GLODAPv2-based C_{ant} section where GLODAPv2 profiles have been co-located with Argo profiles and GLODAPv2 data have been scaled to 2016.

Furthermore, GLODAPv2 gridded product is an annual mean and doesn't resolve the seasonal cycle. It also relies only on ship-based measurements that are biased toward summer (Fig. B top panel). In contrast, Fig. 4 of the manuscript shows that the C_{ant} seasonal cycle in the Irminger Sea (e.g. region 3) nicely follows the seasonal evolution of the MLD (Fig. B bottom panel).

Fig B: Top panel: Monthly distribution of DIC measurements in the GLODAPv2 database (Lauvset et al., 2016) between 2013 and 2018. We selected only the data located in the subpolar North Atlantic gyre (47-65°N; 15-65°W). Bottom panel: Zoom on the C_{ant} section of float 6901026 showing a seasonal cycle in the Irminger Sea (region 3). The black line represents the mixed layer depth. The text was modified as follows:

“in areas where water masses move laterally due to mesoscale processes or circulation changes, gridded climatology such as GLODAPv2 (Lauvset et al., 2016) cannot capture the implied changes in carbon variables. This is because this climatology represents the mean field in which the variability has been smoothed out. However, neural networks, such as ESPER_NN and CONTENT, which derive biogeochemical variables based on water mass characteristics, can cope with these changes. As illustrated on Fig. S7, our Argo-NN-based C_{ant} profiles are in better agreement with ship-based C_{ant} estimates than with GLODAPv2-based C_{ant} profiles.”

Also, the method of calculation of anthropogenic carbon and the equations used are missing.

The equation to compute anthropogenic carbon was added in the supplementary materials.

For example, the study used the nominal year 2015 for their analysis but didn't provide enough detail as to why.

In the revised manuscript, the nominal year 2015 is only used for Fig. 1 where we scaled the GLODAPv2 data via the exponential equation of Carter et al. (2021) to 2015 because it is the central year of our study period (2013-2018). Note that we don't use the nominal year 2015 anymore in subsequent analyses.

Even though the previous literature is cited, the equation should be provided in the supplementary material.

To calculate C_{ant} , we use the ϕC^O_T method, which was specially developed for the Atlantic Ocean. Following your recommendation, we added the main equation of this method in supplementary materials. We also included the reference to Vazquez-Rodriguez et al. (2009) where more details on the method and the full set of equations can be found.

Line 24: Authors claim C_{ant} distribution is well documented but still go ahead to say the deep distribution remains largely unresolved. Please rephrase to avoid confusion for readers.

We rephrase with the sentence :

“Even though the average C_{ant} spatial distribution in the surface North Atlantic is fairly well documented, the deep C_{ant} pathways and their spatio-temporal variability remain largely unresolved.”

Line 25: Argo data and NN? Neural network observation was critical to your findings. This should be stated in your abstract.

We rephrase with the sentence:

“Here we use Argo- O_2 data (pressure, temperature, salinity and oxygen) of 3 floats, spanning the period 2013-2018, as inputs of neural networks to determine macronutrients, total alkalinity and dissolved inorganic carbon. Then we use these two sets of variables to estimate C_{ant} via the back calculation ϕC^O_T method.”

Line 29: Remove the hyphen in high- C_{ant} . You also need to be more explicit when using the word “high.” There is a need to give a reference so that readers have an idea of the C_{ant} concentration or magnitude of the surrounding water with respect to the west of the Reykjanes ridge.

We consider that high C_{ant} values means values higher than 50 $\mu\text{mol/kg}$. We added this specific value in the abstract.

Line 33: I am concerned about the choice of your word “For the first time.” The method is reliable, but it isn’t new (carbon-based back-calculation method by Pérez et al. (2008) and Vazquez-Rodriguez et al. (2009), and it is not the first time work has been done in the North Atlantic (Davila et al., 2022). Using new combined data observations (Argo float and NN) differs from the method itself (which isn’t new).

As also suggested by reviewer #3, we removed the words “For the first time”.

We rephrase this sentence as:

“The method-workflow presented in this “proof-of-concept” study opens up new ways to study the oceanic C_{ant} content at a higher spatio-temporal resolution than shipborne, with further improvements expected with the use of nitrate and pH sensors from the biogeochemical Argo network.”

Line 52: Again, be careful with using verbs like “fast and slow” when used without context or reference. I understand the point you are trying to make, but maybe using a word like “atmospheric $p\text{CO}_2$ has steadily increased over the years since the preindustrial era compared to the oceanic $p\text{CO}_2$. You can go further to quote a figure and provide a reference.

We thank the reviewer for this comment and rephrase with the sentence:

“This net C_{ant} uptake occurs via air-sea exchange, driven by air-sea CO_2 disequilibria resulting from the difference between the steady increase in atmospheric $p\text{CO}_2$ over the years since the pre-industrial era and oceanic $p\text{CO}_2$ (Sabine et al., 2004).”

Line 53: I understand what you were trying to write, but it should be clear that seawater $p\text{CO}_2$ is primarily regulated by seasonal variations in temperature (in the subtropics), whereas at higher latitudes, its oscillation is typically dominated by biological processes (e.g., photosynthetic CO_2 fixation and the remineralization of organic carbon) (Takahashi et al., 2009; Ko et al., 2021). DIC and TA are two of the four marine carbonate systems (the other two are pH and $p\text{CO}_2$) which we use to constrain the carbon cycle. A change in these two will definitely change $p\text{CO}_2$.

Takahashi, T., Sutherland, S. C., Wanninkhof, R., Sweeney, C., Feely, R. A., Chipman, D. W., et al. (2009). Climatological mean and decadal change in surface ocean $p\text{CO}_2$, and net sea-air CO_2 flux over the global oceans. *Deep Sea Res. Pt. II* 56, 554–577. doi: 10.1016/j.dsr2.2008.12.009

Ko Y, Park G-H, Kim D and Kim T-W (2021) Variations in Seawater $p\text{CO}_2$ Associated With Vertical Mixing During Tropical Cyclone Season in the Northwestern Subtropical Pacific Ocean. *Front. Mar. Sci.* 8:679314. doi: 10.3389/fmars.2021.679314

We thank the reviewer for this clarifying statement, which we have integrated as part of the rephrased paragraph:

“The latter is primarily regulated by seasonal variations in temperature in the subtropics whereas at higher latitudes, its oscillation is typically dominated by biological processes (Takahashi et al., 2009) affecting dissolved inorganic carbon (DIC) and total alkalinity (A_T).”

Line 55: For clarity, you should link how the high (low) surface AT (DIC) concentration relates to the low seawater pCO_2 and the subsequent uptake of atmospheric pCO_2 . Provide references where necessary.

Based on a reference suggested by reviewer #3, we rephrased the sentence as:

“A low DIC/ A_T ratio will lead to a large CO_2 uptake capability by the ocean in response to increase in atmospheric pCO_2 (Egleston et al., 2010)”

Line 60-61: The global Cant distribution in the ocean is not homogeneous, neither vertically nor horizontally. Reference?

We added the reference of Davila et al. (2022).

Line 106: Cross-check again. The reference year in both Carter et al. (2021) and Lauvset et al. (2016) is 2002. Where is 2015 from?

We agree that originally the reference year in both Carter et al. (2021) and Lauvset et al. (2016) is 2002. However, to be consistent with our study period (2013-2018) we scaled the data of Lauvset et al. (2016) to the nominal year 2015, with the exponential equation of Carter et al. (2021) (see their Eq. 1), because it is the central year of our study period. In our case, the exponential equation is:

$$C_{ant_2015} = C_{ant_2002} * \exp(0.018989(2015 - 2002)).$$

The scaling equation has been added in the supplementary materials of the revised manuscript.

Line 114: I recommend you add a figure showing the three floats used and their trajectories represented in different colors.

We added a figure in supplementary materials as suggested by the reviewer.

Line 120: Support with Table S1

Done

Line 135: Provide the biases and errors for the nutrients from the NN referred to here.

In the revised manuscript we refer to Table S1, where the uncertainties of the nutrients from the NNs are reported. They represent on average an uncertainty of 2% for the macronutrients and 0.5% for DIC and AT.

Line 136: Instead of referring readers back and forth between papers, you should provide the equations used in the supplementary materials.

We apologize, the term “equation 7” was misleading and we clarified the text as follows: “ESPER_NN was used to obtain the macronutrients (phosphate, nitrate and silicate) using P, T, S, O₂, location and time as predictors.”

Line Line 179-181: Looking at the figures from Line 563 (E.D Fig 4), “the negative $C_{ant,def}$ values (blue color) indicate that this particular parcel has a deficit of C_{ant} and is able to uptake C_{ant} from the atmosphere”. State what the positive values (orange color) represent too. Also, for example, in Fig 4C, there are more negative $C_{ant,def}$ values at depths below compared to around the surface to 800 dbar.

In the revised version of the manuscript, we modified the $C_{ant,def}$ computation. Previously, we used the reference value ($\langle C_{ant} \rangle_{\sigma_{\theta} < 26.5}$) of Ridge & McKinley (2020). This value has been calculated along the segment of the WOCE A22 section between Bermuda and Woods Hole (34-40°N), a region at the northern boundary of the subtropical North Atlantic gyre. Consequently, this reference value might underestimate the C_{ant} concentration at the surface of the subtropics. We computed a new reference value for 2002 with the GLODAPv2 dataset (Lauvset et al., 2016) considering the averaged C_{ant} concentration in surface waters ($\sigma_{\theta} < 26.5 \text{ kg/m}^3$) offshore the Venezuelan coasts (10-20°N; 70-64°W) and found $\langle C_{ant} \rangle_{\sigma_{\theta} < 26.5}^{2002} = 51.5 \text{ } \mu\text{mol/kg}$. We assume that our “reference water mass” offshore the Venezuelan coasts take 5 years to arrive in the Iceland and Irminger basins (Messias & Mercier, 2022). For a given float, we use that reference value rescaled in time to 5 years before the time of its first profile, using the exponential equation of Gruber et al. (2019). For instance, the first profile of the float 6901026, located in the Iceland basin, is in 2012. For this float, the reference value is scaled to the nominal year 2007. Assuming that due to its lagrangian behavior, the float follows the transformation of the same water mass, we kept the same reference value along the float trajectory within the subpolar gyre. However, this assumption is not valid for float 5904988 when it drifted from region 4 to region 1 and entered the North Atlantic Current, where we re-initialised the reference value.

We added these explanations in the “Methods” section of the revised manuscript.

On the revised figures (Fig. S5), the $C_{ant,def}$ values at depth are lower than the values at the surface. This is due to the direct contact between the atmosphere and the ocean surface, where C_{ant} is exchanged.

Does that mean the deep can still take up as much as -15 mol/kg? State clearly in the text where this is first mentioned.

We agree with the reviewer and added the sentence:

“Usually, $C_{ant,def}$ becomes more and more negative with depth (Fig. S5), meaning that the deep waters would uptake a higher amount of C_{ant} compared to the subsurface waters, if they were transported to the ocean surface.”

Line 193: Reading up to this line, I see the mix up at line 106. Your data spans from July 2012 to April 2018; why did you choose the nominal year 2015?

As detailed previously, the reference value for the calculation of $C_{ant,def}$ is now based on the date of the first profile of the float.

Line 214: intermediate depth. State the depth range.

We meant between 400-2000 dbar and this information has been added to the revised manuscript.

Line 223-225: Quote figure to support.

Done

Line 277: Authors need to give a detailed highlight of the role of ocean circulation (not just mention “Daniault et al., 2016”) in the transport of C_{ant} from the surface to the deep waters and the implication for the carbon cycle in the event of climate change (enhanced/reduced circulation).

We rephrased by:

“As inferred by the progression of the float, the C_{ant} -loaded NACW is transported northwards by the NAC (Fig. 1), towards the Iceland Basin (region 2), following the general circulation pattern of the subpolar North Atlantic gyre (Daniault et al., 2016).”

We added a couple of sentences in the “Summary and global implications” of the revised manuscript where we speculate on the implication of an enhanced/reduced oceanic circulation on the distribution of C_{ant} . The sentences are:

“As already demonstrated by different studies (Pérez et al., 2013; Zunino et al., 2014), the AMOC variability affects the northward C_{ant} transport to the subpolar North Atlantic gyre and its storage rate. Under current climate change, the AMOC is projected to weaken, and we could thus expect a concomitant decrease in C_{ant} storage rate and content in the subpolar North Atlantic gyre. Nevertheless, as shown by Brown et al. (2021) and Zunino et al. (2014), the northward oceanic C_{ant} transport might still be subject to continuous increase in response to the rise in anthropogenic CO_2 emissions. Yet, observations on a finer spatio-temporal scale should be sustained in the long-term to determine which effect will be dominant in the future.”

Line 306 and 307: Consistency with using Fig. and Figs when highlighting more than one figure. (Revise entire manuscript).

Done

Line 311: Despite the data spanning five years (July 2012 to April 2018). One significant interpretation missing from this study is the interannual variability (not even mentioned once) and possible implications for the future of the C_{ant} and carbon cycle.

As pointed out by reviewer #3, this study describes a snapshot of the C_{ant} distribution between 2013-2018 and temporal dynamics are not resolved. Our data cannot be used to demonstrate an interannual C_{ant} variability. For instance, for float 6901023, we cannot distinguish that the deeper penetration of C_{ant} in winter 2015 compared to winter 2013 is due to a temporal variability or a spatial variability.

As detailed previously, we added several sentences where we speculate on the potential effect of an AMOC weakening on the future of C_{ant} .

Line 318: “In this region $C_{ant,def} < 5.0 \pm 7.6 \mu\text{mol/kg}$, a value which stays within the method uncertainty (Extended Data Fig. 4)”. Revise the statement. Did you mean $C_{ant,def}$ is below $5.0 \pm 7.6 \mu\text{mol/kg}$?

We revised the sentence accordingly.

Line 436: Add “profiling float” to Argo observation.

Done

Line 436-437: Highlight the potential feedback mechanisms between the deep transport of C_{ant} and climate change, and how will this impact the ocean's ability to absorb and store anthropogenic carbon in the future?

As previously mentioned, we added a comment on this matter:

“As already demonstrated by different studies (Pérez et al., 2013; Zunino et al., 2014), the AMOC variability affects the northward C_{ant} transport to the subpolar North Atlantic gyre and its storage rate. Under current climate change, the AMOC is projected to weaken, and we could thus expect a concomitant decrease in C_{ant} storage rate and content in the subpolar North Atlantic gyre. Nevertheless, as shown by Brown et al. (2021) and Zunino et al. (2014), the northward oceanic C_{ant} transport might still be subject to continuous increase in response to the rise in anthropogenic CO_2 emissions. Yet, observations on a finer spatio-temporal scale should be sustained in the long-term to determine which effect will be dominant in the future.”

Line 443: The methodology utilized data from three different sources Argo float (P, T, S, O₂), ESPER_NN (Nutrients), and CONTENT (DIC and AT), and this contributed to uncertainties that would have been reduced if the data was from a singular insitu source or platform. Highlight the importance of reducing uncertainty for future studies by using a more wholesome platform like the Biogeochemical Argo, which in addition to the P, T, S, and O₂, also has nitrate and pH, from which DIC and AT can be estimated with far less uncertainty. Consider citing the relevant literature.

Claustre, H., Johnson, K.S., Takeshita, Y., 2020. Observing the Global Ocean with Biogeochemical-Argo. *Annu. Rev. Mar. Sci.* 12, 23–48. <https://doi.org/10.1146/annurev-marine-010419-010956>

Addey, C.I. Using Biogeochemical Argo floats to understand ocean carbon and oxygen dynamics. *Nat Rev Earth Environ* 3, 739 (2022). <https://doi.org/10.1038/s43017-022-00341-5>

We agree with the reviewer. Our study focuses on Argo-O₂ because it currently represents the largest database available and wanted to demonstrate the potential of Argo-O₂ array for studying C_{ant}. Based on that, we elucidated the C_{ant} pathways towards the deep North Atlantic Ocean. For the future, the BGC-Argo mission plans to maintain in operation about 1000 BGC-Argo floats equipped with O₂ and pH sensors. Yet, many additional Argo-O₂ only floats will be deployed, resulting therefore in a larger Argo-O₂ array compared to the BGC one (see <https://fleetmonitoring.euro-argo.eu/dashboard?Status=Active>). The idea is thus to use our methodology to provide C_{ant} estimates at a better resolution than the one provided by BGC-Argo. Scientific studies based on both BGC-Argo (nitrate and pH sensors), used to train the NN, and Argo-O₂ data will certainly have reduced uncertainties.

In the “Summary and global implications” we added the sentences:

“In addition, the methodological uncertainty would be reduced by using data from the growing BGC-Argo database (Addey, 2022; Claustre et al., 2020) to train the neural networks. Although floats from the BGC-Argo programme, equipped with pH and nitrate sensors, were not included in our study because available floats did not follow the main circulation patterns of the subpolar North Atlantic gyre, BGC-Argo floats can also be used to derive C_{ant} estimates.”

Lastly, the authors heavily relied on the NN estimates for their analysis, but this isn't captured in the manuscript title and keywords (only mentions Argo-O₂ floats). Authors should revise the manuscript title to reflect the combined Argo float and NN approach.

We changed the title by: “Anthropogenic carbon pathways towards the deep North Atlantic revealed by Argo-O₂ data combined with neural networks and back calculations”

We also changed the key words.

REFERENCES

Bittig, H.C., et al. "An alternative to static climatologies: Robust estimation of open ocean CO₂ variables and nutrient concentrations from T, S, and O₂ data using Bayesian neural networks." *Frontiers in Marine Science* 5 (2018): 328.

Davila, X., et al. "How is the ocean anthropogenic carbon reservoir filled?." *Global Biogeochemical Cycles* 36.5 (2022): e2021GB007055.

Egleston, E., et al. "Revelle revisited: Buffer factors that quantify the response of ocean chemistry to changes in DIC and alkalinity." *Global Biogeochemical Cycles* 24.1 (2010).

Lauvset, S.K., et al. "A new global interior ocean mapped climatology: The 1× 1 GLODAP version 2." *Earth System Science Data* 8.2 (2016): 325-340.

Ridge, S. M., and G. A. McKinley. "Advective controls on the North Atlantic anthropogenic carbon sink." *Global Biogeochemical Cycles* 34.7 (2020): e2019GB006457.

Vázquez-Rodríguez, M., et al. "An upgraded carbon-based method to estimate the anthropogenic fraction of dissolved CO₂ in the Atlantic Ocean." *Biogeosciences Discussions* 6.2 (2009): 4527-4571.

Zunino, P., et al. "Variability of the transport of anthropogenic CO₂ at the Greenland–Portugal OVIDE section: controlling mechanisms." *Biogeosciences* 11.8 (2014): 2375-2389.

Reviewer #3 (Remarks to the Author):

Dear Rémy Asselot and colleagues,

It was a pleasure to read and think about your manuscript entitled “Argo-O₂ floats reveal the anthropogenic carbon pathways towards the deep North Atlantic”, in which the ocean interior distribution of anthropogenic carbon (Cant) is derived from oxygen, salinity, temperature and pressure measurement obtained from three Argo floats that travelled through major parts of the North Atlantic Ocean from 2013 through 2018. The reconstructed patterns in Cant are further related to the distribution of water masses and previous knowledge about their transport, such that transport processes of Cant can be inferred.

I consider this study very timely and important, given the current challenge of the ocean carbon community to resolve ocean interior carbon dynamics at high resolution. This challenge appears particularly relevant when considering the rapid changes in the ocean carbon cycle expected under declining CO₂ emissions and the potential implementation of carbon dioxide removal activities. Your study has the potential to contribute significantly to tackling this challenge. It is mostly transparently described and provides - with few exceptions - sufficient methodological details. The results appear solid as presented, the text is well written and structured, and figures are clear and informative.

However, a few (interconnected) points may require major revisions to ensure the correct interpretation and contextualisation of the results, which is important for the community to derive precise conclusions concerning the future application of this method and inherent limitations. I organised my main concerns into four sections below, while detailed comments are provided in the attached pdf file.

We would like to thank Jens Daniel Müller for his valuable insights and comments. We are grateful for your effort reviewing our paper. Here we answer the main concerns of the reviewer while we answer the detailed comments directly in the PDF file.

Data selection

This study is based on a selection of three Argo floats providing observations from the North Atlantic between 2013 - 2018. Thus, the results provide a snapshot of the Cant distribution during this period, while temporal dynamics are not resolved in this study. My impression is

that this nature of the results needs to be expressed more clearly in the abstract to avoid raising false expectations.

We fully agree and in the abstract, we added the sentence:

“Here we use Argo-O₂ data (pressure, temperature, salinity and oxygen) of 3 floats, spanning the period 2013-2018, as inputs of neural networks to determine macronutrients, total alkalinity and dissolved inorganic carbon. Then we use these two sets of variables to estimate C_{ant} via the back calculation ϕC^O_T method.”

Given that the float data as used in this study allows only for a snapshot in time, I was wondering about the added value of using the float data and not a static climatology of the input variables, which would permit resolving spatial patterns over a three-dimensional grid covering the whole study region. Maybe the advantages of using the float data could be described more clearly to motivate this study.

With Argo-O₂ float, combined with neural networks, we obtain data covering the whole year with one C_{ant} profile every 10 days. Annual mean climatology, such as GLODAPv2 (Lauvset al., 2016), are based on ship-based measurements that are seasonality biased because most cruises take place in summer months (see Fig. 2 in our answer to reviewer #2). The main advantage of Argo data is that they are seasonally unbiased, covering all seasons, and thus they have a higher temporal resolution (see the answer to the major comment of reviewer #2).

We added a couple of sentences in the “Methods” part:

“in areas where water masses move laterally due to mesoscale processes or circulation changes, gridded climatology such as GLODAPv2 (Lauvset et al., 2016) cannot capture the implied changes in carbon variables. This is because this climatology represents the mean field in which the variability has been smoothed out. However, neural networks, such as ESPER_NN and CONTENT, which derive biogeochemical variables based on water mass characteristics, can cope with these changes. As illustrated on Fig. S7, our Argo-NN-based C_{ant} profiles are in better agreement with ship-based C_{ant} estimates than with GLODAPv2-based C_{ant} profiles.”

In this regard, I deem it also important to inform the reader about the criteria based on which the three floats were selected. Specifically, you may want to address why this study does not use the full fleet of Argo-O₂ profiles from this region in order to gain a more complete picture of the C_{ant} distribution in space.

The scope of the study is to investigate which are the pathways followed by C_{ant} to penetrate the deep levels of the subpolar North Atlantic gyre. We selected **all** Argo floats that have a long life time (more than 3 years) and that follow the main cyclonic circulation of the subpolar North Atlantic gyre. Even if our database contains only 3 Argo-O₂ floats, these 3 floats bring valuable information on the C_{ant} pathways. We added the sentences:

“We selected all Argo-O₂ floats following a cyclonic pathway in the subpolar gyre of the North Atlantic Ocean (Fig. S1) and that have a life longer than 3 years. With these criteria, our dataset is composed of 3 floats.”

In addition, I was surprised that this study focuses exclusively on O₂ as a biogeochemical input parameter for the determination of the CO₂ system and - based on that - the C_{ant}

concentration. Wouldn't float-based pH measurement provide a valuable (additional) constraint on the CO₂ system? I imagine that the availability of float pH data in the study region might not be sufficient to base the results of this study on pH measurements. Nevertheless, it appears important to address the potential use of pH measurements to guide future Argo deployments and research on the topic.

As explained to reviewer #2, our study focuses on Argo-O₂ because it currently represents the largest database available and wanted to demonstrate the potential of Argo-O₂ array for studying C_{ant}. Based on that, we elucidated the C_{ant} pathways towards the deep North Atlantic Ocean. For the future, the BGC-Argo mission plans to maintain in operation about 1000 BGC-Argo floats equipped with O₂ and pH sensors. Yet, many additional Argo-O₂ only floats will be deployed, resulting therefore in a larger Argo-O₂ array compared to the BGC one (see <https://fleetmonitoring.euro-argo.eu/dashboard?Status=Active>). The idea is thus to use our methodology to provide C_{ant} estimates at a better resolution than the one provided by BGC-Argo. Scientific studies based on both BGC-Argo (nitrate and pH sensors, used to train the NN, and Argo-O₂ data will certainly have reduced uncertainties.

In the "Summary and global implications" we added the sentences:

"In addition, the methodological uncertainty would be reduced by using data from the growing BGC-Argo database (Addey, 2022; Claustre et al., 2020) to train the neural networks. Although floats from the BGC-Argo programme, equipped with pH and nitrate sensors, were not included in our study because available floats did not follow the main circulation patterns of the subpolar North Atlantic gyre, BGC-Argo floats can also be used to derive C_{ant} estimates."

Uncertainty estimates

A bulk uncertainty of the C_{ant} estimates ($\pm 7.6 \mu\text{mol/kg}$) is provided that is based on a Monte Carlo approach obtained from varying the input parameters of the C_{ant} calculation within their ranges of uncertainty. If my understanding of this procedure is correct, then the primary outcome should be an individual uncertainty estimate associated with each calculated C_{ant} value. These individual uncertainty estimates should vary with the uncertainty of the input parameters. This understanding evokes a couple of questions: (1) Is it justified and appropriate to average the individual uncertainties into a single bulk value that is then reported identically for each C_{ant} estimate? My impression is that it would be more informative to provide specific uncertainty estimates for each C_{ant} value, such that one can distinguish results with higher and lower confidence.

We thank the reviewer for this interesting point that improves the uncertainty analysis. It is exact that the primary outcome of our Monte Carlo analysis is an individual uncertainty associated with each C_{ant} value. As suggested by the reviewer, we changed our approach and rather than giving a bulk C_{ant} uncertainty reported for each C_{ant} estimate, we provide specific C_{ant} uncertainty for each C_{ant} value. Thanks to the reviewer comment, we also revised our uncertainty analysis. We realized that, previously, the uncertainty of the $\phi\text{C}^{\text{O}_T}$ method was counted twice, overestimating the uncertainty of our methodological approach (Argo + NN + $\phi\text{C}^{\text{O}_T}$ method). The revised uncertainty is still computed via a Monte Carlo analysis, fluctuating between $\pm 5.4 \mu\text{mol/kg}$ and $\pm 10.2 \mu\text{mol/kg}$ and has an average uncertainty of $\pm 5.9 \mu\text{mol/kg}$.

These values are added in the revised manuscript.

(2) Does the uncertainty estimate include a specific contribution that arises from CO₂ system calculations? In this regard it would also be informative to know how the ϕC°_T method differs when it is provided with AT and DIC (TSO2-NN approach) or AT and pH (standard approach) data. Please note that this comment links also to my first general comment regarding the use of directly measured Argo pH data.

When using A_T and pH directly from observations (standard approach), the ϕC°_T method gives an averaged uncertainty of $\pm 5.2 \mu\text{mol/kg}$ (Vázquez-Rodríguez et al., 2009). Our methodology, using A_T and DIC from neural networks (TSO2-NN approach) gives an overall averaged uncertainty of $\pm 5.9 \mu\text{mol/kg}$. This overall methodological uncertainty is close to the original uncertainty of the ϕC°_T method, meaning that most of the uncertainty comes from the ϕC°_T method. On Fig. S7 we actually compared the standard approach (blue profiles) with the TSO2-NN approach (red profiles and black profiles) where the uncertainties of the two approaches are reported on the figures.

We agree that, in the future, BGC-Argo floats equipped with pH sensors could be used for the training phase of neural networks, reducing the uncertainty on the C_{ant} estimates.

(3) According to table S1, the uncertainty of DIC as an input parameter is $>10 \mu\text{mol/kg}$ for all three floats. How is it possible that the uncertainty of the derived C_{ant} estimate is lower than that uncertainty of the input parameter?

The C_{ant} concentrations are driven by DIC and O_2 concentrations. These two variables are anti-correlated with, usually, more O_2 generates less DIC (see figure below) due to several biogeophysical processes (Louanchi et al., 2001). Since DIC decreases when O_2 increases, there is a negative correlation between DIC and O_2 uncertainties, meaning that DIC uncertainties are partially compensated by O_2 uncertainties in the computation of C_{ant} estimates. This compensation explains the lower C_{ant} uncertainties compared to DIC uncertainties. We add this explanation in the supplementary materials of the revised manuscript.

Relation between DIC concentrations and O_2 concentrations for the Argo float 6901023.

Applicability of the method to achieve unprecedented temporal resolution of the C_{ant} propagation into the ocean interior

While the methods used in this study allows to provide a snapshot of the C_{ant} distribution in the 2010s and link this distribution to the presence of water masses and their transport, I'm doubtful if it will enable us to substantially increase the temporal resolution with which we can track the propagation of C_{ant} into the ocean interior. My main concern in this regard arises from the uncertainty inherent to the determination of C_{ant} . Assuming that the uncertainty estimate provided in this study ($\pm 7.6 \mu\text{mol/kg}$) is correct, then the uncertainty is of similar magnitude as the decadal changes in the concentration of C_{ant} that the upper ocean experiences at current (still high!) rates of increase in atmospheric CO_2 . It is thus questionable whether sub-decadal changes in the storage of C_{ant} can be resolved with the method as presented in this study. Given that decadal-scale changes in the C_{ant} storage can be determined from ship-based measurements as well, I encourage the authors to revise their conclusions that "unprecedented" temporal resolution can be achieved and provide a more thorough assessment of this anticipated skill.

We agree that the methodological uncertainty is of similar magnitude as the decadal changes in surface C_{ant} concentration. Our methodological approach that combines Argo- O_2 data with neural networks to study C_{ant} needs to be improved in the future to reduce this uncertainty (e.g. decrease the uncertainty of the ϕC^{O_T} method or deployment of Argo floats with pH sensors to improve the uncertainties of neural networks). With the revised uncertainty analysis, the overall averaged uncertainty of $\pm 5.9 \mu\text{mol/kg}$ represents the standard error for a given estimate along a float trajectory. The standard error will be smaller when considering inventories obtained by averaging several Argo profiles in a given box (e.g. $2^\circ \times 2^\circ$) over a month because this random uncertainty decreases as the number of Argo floats used in the estimate increases.

Argo floats data cover the whole year and provide one C_{ant} profile every 10 days. These data are not seasonality biased compared to ship-based measurements that often have observations covering the summer months (see answer reviewer #2). In this way, we see our methodological approach as a method that increase C_{ant} estimates at an "unprecedented" temporal resolution. In addition, our method permits to detect the effect of mesoscale features, such as eddies, on C_{ant} distribution. With ship-based measurements, these original results would have not been possible, demonstrating the increased temporal resolution with Argo float observations.

I'm further wondering how convinced the authors are of the skill of their method to separate changes in anthropogenic and natural DIC. Imagining for example that the surface ocean was losing natural DIC and oxygen due to warming, would you expect that two subsequent C_{ant} estimates obtained with the ϕC^{O_T} method would be perturbed by co-occurring changes in these variables or not?

Our study is based on the ϕC^{O_T} method which is a back-calculation method and where natural DIC and oxygen are input variables. As all the back-calculation methods, the ϕC^{O_T} method assumes that:

$$C_{ant} = \text{DIC}_{\text{natural}} - \text{DIC}_{\text{bio}} - \text{DIC}_{\text{preformed}}$$

Where $\text{DIC}_{\text{natural}}$ represents the natural DIC pool, DIC_{bio} represents the DIC generated by biological activity (e.g. remineralization) and $\text{DIC}_{\text{preformed}}$ represents the preformed DIC. As

demonstrated by this equation, if natural DIC decreases, C_{ant} concentration will also be reduced.

In the equation above, the DIC_{bio} is assumed to be:

$$\text{DIC}_{\text{bio}} = \text{AOU}/R_c$$

Where AOU is the apparent oxygen utilization (oxygen saturation minus observed oxygen concentration under the same temperature and salinity) and R_c is the Redfield ratio $\text{O}_2:\text{C}$. A decrease in oxygen would lead to higher AOU, indicating a higher biological activity. As a consequence, the DIC_{bio} would increase and thus C_{ant} concentration would be reduced.

Emphasis on eddy pumping

I was fascinated by the co-location of anticyclonic eddies and the deep extension of high C_{ant} concentrations identified in this study. It is a prime example how the frequent and repeated Argo observations help to better understand the cycling of carbon in the ocean interior.

Indeed, the frequent and repeated Argo observations help to better understand the carbon cycle, that is what we meant when we talk about increasing the temporal resolution of C_{ant} estimates at an unprecedented scale.

As this topic is currently restricted to a few sentences in the results and discussion, I encourage the authors to put a bit more emphasis on these exciting findings. Some lead questions that might be worth considering when expanding on this topic include: Are all anticyclonic eddies detected along the float trajectories associated with a deep penetration of C_{ant} , or are some eddies not effective? How does the strength of the deep penetration signal relate to the uncertainty of the C_{ant} estimate? Can it be excluded that the deep C_{ant} penetration is an artefact of changes in the O_2 distribution induced by the eddies?

We agree that the exciting results need further investigations but we prefer to limit ourselves to explain the physics behind the deep C_{ant} pulses.

Our data indicate that the deep penetration of C_{ant} generated by eddies does not affect the uncertainty of C_{ant} estimates. The uncertainties remain similar between the Argo profiles inside and outside of an anticyclonic eddy. For instance, for the Argo float 5904988, we isolate the profile occurring on the 10th of June, 2016 and located in an anticyclonic eddy. Its overall uncertainty along the whole profile is $\pm 6.9 \mu\text{mol}/\text{kg}$, within the range of uncertainties (± 5.4 to $\pm 10.2 \mu\text{mol}/\text{kg}$).

Eddies show strong isopycnal slopes due to the geostrophic balance. However, when looking along an isopycn, eddies have the same tracer content (e.g. T, S, O_2 and inferred C_{ant}) as their formation region at the same density so that there is no artifact linked to changes in O_2 associated with eddies. The eddies carry these properties away from their formation regions and by doing so contribute to the isopycnal mixing of tracers. The downward isopycnal displacement generated by the anticyclonic eddies does not correspond to a diapycnal downwelling.

We revised manuscript by adding the sentences:

“In the WBC, we observe the occurrence of occasional high- C_{ant} pulses throughout the water column (black arrows on Figs. 2b and 3b) that we relate to anticyclonic mesoscale eddies (Fig. S6). These anticyclonic features lead to a punctual downward isopycnal displacement

of surface waters containing high C_{ant} concentration, explaining the C_{ant} pulses identified in our sections. Such eddies could carry C_{ant} away from their formation regions and when they collapse, they might contribute to the isopycnal mixing of this tracer. ”

I hope you find this feedback helpful to improve some aspects of your study. Please do not hesitate to get in touch if any comments are not clear to you.

We thank you again for your valuable suggestions that helped to greatly improve the manuscript.

Best wishes
Jens Daniel Müller

REFERENCES

Lauvset, S.K., et al. "A new global interior ocean mapped climatology: The 1× 1 GLODAP version 2." *Earth System Science Data* 8.2 (2016): 325-340.

Louanchi, F., et al. "Dissolved inorganic carbon, alkalinity, nutrient and oxygen seasonal and interannual variations at the Antarctic Ocean JGOFS-KERFIX site." *Deep Sea Research Part I: Oceanographic Research Papers* 48.7 (2001): 1581-1603.

Vázquez-Rodríguez, M., et al. "An upgraded carbon-based method to estimate the anthropogenic fraction of dissolved CO₂ in the Atlantic Ocean." *Biogeosciences Discussions* 6.2 (2009): 4527-4571.

REVIEWER COMMENTS

Reviewer #2 (Remarks to the Author):

Dear Asselot et,

I am satisfied with your reply to my comments. One last observation I made was the use of kg/m³ and kg m⁻³. For consistency, stick to one format. On most figures, you have kg/m³. Whichever you decide to use, just make sure it is consistent.

Reviewer #4 (Remarks to the Author):

Summary and overall Impression

The authors utilize data from three BGC-Argo floats of oxygen, temperature, salinity, and pressure over 5 years in the North Atlantic, in combination with existing neural network and back propagation approaches. With this approach, they estimate anthropogenic carbon (C_{ant}) and investigate its pathways from the surface to the deep North Atlantic. The authors claim that this method allows for an investigation of C_{ant} at higher spatio-temporal resolution than was previously possible with ship-based data. The authors then relate the inferred C_{ant} patterns to the water masses to highlight the role of water mass transformation along the pathways of the subpolar gyre as an important mechanism for C_{ant} penetration at depth

The paper will be of high interest, especially for the ocean carbon cycle and BGC-Argo float communities. The results appear robust, and the interpretations and conclusions are supported by the results. The figures are clear and informative as well. However, I have major concerns that I believe should be addressed before publication, as well as some minor suggestions to improve clarity, although the text is otherwise well-written. I recommend publication after these revisions have been addressed.

Major Comments

It is not entirely clear to me what the main goal of the study is, and if the chosen method is the more appropriate for this goal.

One goal could be to better understand the connection between the C_{ant} distribution and the water masses. If so, would it not make more sense, to look at the C_{ant} and water mass distribution using existing climatologies of T, S, and O₂ and calculating C_{ant} using the analog method? With such C_{ant} estimates, one would be able to connect the penetration of C_{ant} with the different water masses. A similar point was raised by reviewer #2 in the previous round. In their response, the authors had argued that the climatologies do not resolve mesoscale features and the lateral movement of water masses, while the BGC-Argo floats are able to capture this. What is the broader impact of these estimates at the mesoscale? Can we really make conclusions on the mesoscale given the uncertainties of the method? This should at least be discussed.

If the goal is to quantify the effect of mesoscale eddies on Cant, then the float profiles should be co-located with eddies, e.g., from Aviso. And again, the authors would need to check if it is possible to detect such mesoscale imprints on Cant within the uncertainties of the method.

Further, the authors state in their response to Reviewer #2 that “GLODAPv2 gridded product is an annual mean and doesn’t resolve the seasonal cycle”. However, we have mapped DIC and alkalinity monthly climatologies based on GLODAP data by Keppler et al., 2020, Brouillon et al., 2020, and Brouillon et al., 2019 so the argument doesn’t hold as those climatologies are available.

In addition, beyond the monthly climatologies, the authors could use the mapped monthly resolved fields of T/S by Roemmich & Gilson 2009, and the mapped monthly resolved O2 fields by Sharp et al., 2023 to calculate Cant with the analog method. The mapped monthly fields of DIC, MOBO-DIC by Keppler et al. 2023, might also be useful. With these mapped data products at monthly resolution, a more representative estimate of the changes in Cant could possibly be achieved (assuming the uncertainties allow such an estimate), compared to the estimate from the 3 floats, and it would be at higher resolution than the decadal eMLRC* estimates.

Another goal could be to say with Argo we can investigate the change in Cant at higher temporal resolution than with ship data, e.g., investigating the Cant increase at least at annual resolution (in order to have an added value from the existing decadal delta Cant estimates with the eMLRC* method). However, the question is again if the uncertainty of the method is small enough to distinguish interannual signals. This may not be possible.

I suggest the authors state the exact goal(s) of this study in the introduction and then communicate clearly why the chosen method is best for this goal. I’m not suggesting disregarding the method altogether. It just should be clearer why this method was chosen and what the goal of this study is, along with what previous gap it is closing.

Minor Comments

1) Structure: many paragraphs are very long e.g., the first paragraph in the Introduction is a whole page. The paper would be a lot clearer and easier to read if each paragraph only dealt with one topic and was thus shorter.

2) The Introduction could be improved:

- It should be clearer what’s new in this study compared to previous estimates. That message is indirectly in the introduction but should be made more explicitly (see also my major comment).

- Reading the introduction, I assumed that the authors had developed a new neural network approach and wondered why they didn’t use existing frameworks such as ESPER, CONTENT, and CANYON. In the Methods section, it becomes clear, that they use those existing frameworks. This should be mentioned in the introduction, even if it is just by adding the words ‘previously developed’ to neural networks (this was done in the abstract with ‘existing neural networks’, but I missed it in the introduction). Consider explicitly mentioning CANYON, CONTENT, and ESPER in the introduction to make it even clearer.

3) I find the Discussion a bit confusing, and it doesn't flow well because it mixes insights from previous studies with insights from this study, and the separation is not always clear. This section could be more powerful if it was more clearly separated (as well as shorter paragraphs, as mentioned above).

Specific Comments

L24: But this study also doesn't address the deep open beyond 2000 m. Consider rephrasing to 'sub-surface' or 'upper ocean' instead of 'deep'

L24: We do know a bit about spatio-temporal variability of Cant in the North Atlantic from Sabine et al., 2004, Gruber et al., 2019, and Mueller et al. 2023. It's just that those studies are on decadal timescales, while this study is looking at shorter timescales.

L35: Can Cant have an amplitude? Consider rephrasing, e.g., to "amount of Cant" if that's what you mean.

L52: Remove 'total' (as the number provided is per year)

L94: Consider adding also the importance of understanding Cant in light of (hopefully) CO2 emission reductions and carbon removal strategies.

L97: Not only the deep, also the upper Cant. Rephrase e.g., to "upper ocean"

L100: Add a reference to this new study: Mueller et al., 2023

L110: Add that 'seasonally unbiased' is compared to ship data

Fig.1: Add that the purple arrows are between the upper and deep

L242: It should be added if the visual inspection ever disagrees with the MLD estimate. And if so, what was done when the visual inspection disagreed with the MLD estimate?

L274: Why was not a more up-to-date method used, e.g., AVISO META by Pegliasco et al. (2022). (It's not a big deal and doesn't need to be changed, I'm just curious).

L296: floats' (not float's)

I hope you find this review helpful. Best wishes and all the best with the publication of this important and exciting manuscript.

References

Broullón, D., Pérez, F. F., Velo, A., Hoppema, M., Olsen, A., Takahashi, T., Key, R. M., Tanhua, T., González-Dávila, M., Jeansson, E., Kozyr, A., & van Heuven, S. M. A. C. (2019). A global monthly climatology of total alkalinity: A neural network approach. *Earth System Science Data*, 11(3), 1109–1127. <https://doi.org/10.5194/essd-11-1109-2019>

Broullón, D., Pérez, F. F., Velo, A., Hoppema, M., Olsen, A., Takahashi, T., Key, R. M., Tanhua, T., Santana-Casiano, J. M., & Kozyr, A. (2020). A global monthly climatology of oceanic total dissolved inorganic

carbon: A neural network approach. *Earth System Science Data*, 12(3), 1725–1743.
<https://doi.org/10.5194/essd-12-1725-2020>

Gruber, N., Landschützer, P., & Lovenduski, N. S. (2019). The Variable Southern Ocean Carbon Sink. *Annu. Rev. Mar. Sci.* 2019.

Keppler, L., Landschützer, P., Gruber, N., Lauvset, S. K., & Stemmler, I. (2020). Seasonal Carbon Dynamics in the Near-Global Ocean. *Global Biogeochemical Cycles*, 34(12), e2020GB006571.
<https://doi.org/10.1029/2020GB006571>

Keppler, L., Landschützer, P., Lauvset, S. K., & Gruber, N. (2023). Recent Trends and Variability in the Oceanic Storage of Dissolved Inorganic Carbon. *Global Biogeochemical Cycles*, 37(5), e2022GB007677.
<https://doi.org/10.1029/2022GB007677>

Müller, J. D., Gruber, N., Carter, B., Feely, R., Ishii, M., Lange, N., Lauvset, S. K., Murata, A., Olsen, A., Pérez, F. F., Sabine, C., Tanhua, T., Wanninkhof, R., & Zhu, D. (2023). Decadal Trends in the Oceanic Storage of Anthropogenic Carbon From 1994 to 2014. *AGU Advances*, 4(4), e2023AV000875.
<https://doi.org/10.1029/2023AV000875>

Pegliasco, C., Delepouille, A., Mason, E., Morrow, R., Faugère, Y., & Dibarboure, G. (2022). META3.1exp: A new global mesoscale eddy trajectory atlas derived from altimetry. *Earth System Science Data*, 14(3), 1087–1107. <https://doi.org/10.5194/essd-14-1087-2022>

Roemmich, D., & Gilson, J. (2009). The 2004-2008 mean and annual cycle of temperature, salinity, and steric height in the global ocean from the Argo Program. *Progress in Oceanography*, 52(2), 81–100.
<https://doi.org/10.1016/j.pocean.2009.03.004>

Sabine, C. L., Feely, R. A., Gruber, N., Key, R. M., Lee, K., Bullister, J. L., Wanninkhof, R., Wong, C. S., Wallace, D. W. R., Tilbrook, B., Millero, F. J., Peng, T. H., Kozyr, A., Ono, T., & Rios, A. F. (2004). The oceanic sink for anthropogenic CO₂. *Science*, 305(5682), 367–371. <https://doi.org/10.1126/science.1097403>

Sharp, J. D., Fassbender, A. J., Carter, B. R., Johnson, G. C., Schultz, C., & Dunne, J. P. (2023). GOBAI-O2: Temporally and spatially resolved fields of ocean interior dissolved oxygen over nearly two decades. *Earth System Science Data Discussions*, 1–46. <https://doi.org/10.5194/essd-2022-308>

REVIEWER COMMENTS

Reviewer #2 (Remarks to the Author):

Dear Asselot et,

I am satisfied with your reply to my comments. One last observation I made was the use of kg/m³ and kg m⁻³. For consistency, stick to one format. On most figures, you have kg/m³. Whichever you decide to use, just make sure it is consistent.

First, we would like to thank the reviewer for their review and their valuable comments. We revised the figures and adopted the format kg m⁻³.

Reviewer #4 (Remarks to the Author):

Summary and overall Impression

The authors utilize data from three BGC-Argo floats of oxygen, temperature, salinity, and pressure over 5 years in the North Atlantic, in combination with existing neural network and back propagation approaches. With this approach, they estimate anthropogenic carbon (C_{ant}) and investigate its pathways from the surface to the deep North Atlantic. The authors claim that this method allows for an investigation of C_{ant} at higher spatio-temporal resolution than was previously possible with ship-based data. The authors then relate the inferred C_{ant} patterns to the water masses to highlight the role of water mass transformation along the pathways of the subpolar gyre as an important mechanism for C_{ant} penetration at depth.

The paper will be of high interest, especially for the ocean carbon cycle and BGC-Argo float communities. The results appear robust, and the interpretations and conclusions are supported by the results. The figures are clear and informative as well. However, I have major concerns that I believe should be addressed before publication, as well as some minor suggestions to improve clarity, although the text is otherwise well-written. I recommend publication after these revisions have been addressed.

We thank the reviewer for their helpful comments and valuable insights that significantly improved our manuscript. We are grateful for their efforts reviewing our paper.

Major Comments

It is not entirely clear to me what the main goal of the study is, and if the chosen method is the more appropriate for this goal.

One goal could be to better understand the connection between the C_{ant} distribution and the water masses. If so, would it not make more sense, to look at the C_{ant} and water mass distribution using existing climatologies of T, S, and O₂ and calculating C_{ant} using the analog method? With

such C_{ant} estimates, one would be able to connect the penetration of C_{ant} with the different water masses. A similar point was raised by reviewer #2 in the previous round. In their response, the authors had argued that the climatologies do not resolve mesoscale features and the lateral movement of water masses, while the BGC-Argo floats are able to capture this. What is the broader impact of these estimates at the mesoscale? Can we really make conclusions on the mesoscale given the uncertainties of the method? This should at least be discussed.

If the goal is to quantify the effect of mesoscale eddies on C_{ant} , then the float profiles should be co-located with eddies, e.g., from Aviso. And again, the authors would need to check if it is possible to detect such mesoscale imprints on C_{ant} within the uncertainties of the method.

Further, the authors state in their response to Reviewer #2 that “GLODAPv2 gridded product is an annual mean and doesn’t resolve the seasonal cycle”. However, we have mapped DIC and alkalinity monthly climatologies based on GLODAP data by Keppler et al., 2020, Broullon et al., 2020, and Broullon et al., 2019 so the argument doesn’t hold as those climatologies are available. In addition, beyond the monthly climatologies, the authors could use the mapped monthly resolved fields of T/S by Roemmich & Gilson 2009, and the mapped monthly resolved O₂ fields by Sharp et al., 2023 to calculate C_{ant} with the analog method. The mapped monthly fields of DIC, MOBO-DIC by Keppler et al. 2023, might also be useful. With these mapped data products at monthly resolution, a more representative estimate of the changes in C_{ant} could possibly be achieved (assuming the uncertainties allow such an estimate), compared to the estimate from the 3 floats, and it would be at higher resolution than the decadal eMLRC* estimates.

Another goal could be to say with Argo we can investigate the change in C_{ant} at higher temporal resolution than with ship data, e.g., investigating the C_{ant} increase at least at annual resolution (in order to have an added value from the existing decadal delta C_{ant} estimates with the eMLRC* method). However, the question is again if the uncertainty of the method is small enough to distinguish interannual signals. This may not be possible.

I suggest the authors state the exact goals(s) of this study in the introduction and then communicate clearly why the chosen method is best for this goal. I’m not suggesting disregarding the method altogether. It just should be clearer why this method was chosen and what the goal of this study is, along with what previous gap it is closing.

The goal of this study is to demonstrate that Argo-O₂ data combined with neural networks and a back calculation method can be used to investigate C_{ant} distribution in the Subpolar North Atlantic. We selected 3 Argo-O₂ floats that: (1) circulated along the subpolar North Atlantic gyre and (2) crossed the A25 OVIDE GO-SHIP line, for which water samples were available to determine, through the same back-calculation method, the C_{ant} concentration. The results revealed a good agreement between the Argo-based and the ship-based C_{ant} estimates. Consequently, taking advantage of the quasi-lagrangian tracking of Argo-O₂ floats, we described the C_{ant} distribution along the pathways of the North Atlantic Central Water into Subpolar Mode Water and Labrador Sea Water. We showed how the C_{ant} distribution is tightly linked to water mass transformation. We also demonstrated a stepwise deepening of C_{ant} along the main oceanic circulation of the

subpolar North Atlantic gyre. As kindly acknowledged by the reviewer, our approach is robust and the results are supported by our data analysis.

We agree that other approaches could have been possible to investigate C_{ant} distribution in the subpolar North Atlantic gyre. One approach is the use of climatologies. The main advantage of climatologies is to provide a gridded view at regional or global scale, which is useful to describe mean spatial patterns. However, every method has its advantages and disadvantages. The main difficulty in using climatologies in this study is that those available do not cover the same time period and thus do not represent the same mean oceanic state. For instance, the monthly fields of DIC, MOBO-DIC (Keppler et al., 2023), have been averaged over 2004-2019 while the monthly climatology of total alkalinity (Brouillon et al., 2019) is averaged over the 1991-2016 period. In addition, the mapped monthly O_2 fields of Sharp et al. (2023) represents an average over the 2004-2021 period while the monthly field of T/S (Roemmich & Gilson, 2009) is averaged over the 2004-2022 period. To avoid the generation of unrealistic patterns, as might be observed when producing T/S climatologies for instance (Lozier et al., 1994), cautious work would have been necessary to combine the different climatologies. Additionally, the climatology-based C_{ant} profiles along the OVIDE line are biased compared to the ship-based C_{ant} data (Supplementary Fig. 7). We believe that using climatology could be an interesting approach but it deserves further investigation. Moreover, it does not satisfy our goal to demonstrate the possibility to infer C_{ant} distribution from Argo- O_2 data, and ultimately its evolution. Additionally, as acknowledged by the reviewer, Argo- O_2 data have a higher spatial resolution than ship-based measurements and could be used in a future work to investigate the interannual to decadal changes of C_{ant} . Our Argo-based C_{ant} estimates assess a C_{ant} uncertainty of $\pm 5.9 \mu\text{mol kg}^{-1}$, which is compatible with the C_{ant} change amplitude higher than $12 \mu\text{mol kg}^{-1}$ in the North Atlantic Ocean over 2004-2014 observed by Müller et al. (2023).

The goal of the study is not to quantify the effect of mesoscale eddies but to explain the C_{ant} distribution along the float pathways. We identified deep C_{ant} pulses that we related to mesoscale eddies. These eddies correspond to a change in isopycnal depth without a first-order change in the properties of the water masses. This exciting result needs further investigation that is beyond the scope of this study. The estimated C_{ant} uncertainty is the same inside and outside the eddies.

We clarified the goal of our study in the introduction, and we added the following sentence: “Considering the unrivaled spatio-temporal sampling provided by the Argo- O_2 network (Roemmich et al., 2019), the purpose of this study is to demonstrate that Argo- O_2 data combined with existing neural networks (i.e., ESPER_NN (Carter et al., 2021), CANYON-B and CONTENT (Bittig, Steinhoff, et al., 2018)) and a back-calculation method can be used to obtain reliable C_{ant} estimate at the finest spatio-temporal scale to date.”

Minor Comments

1) Structure: many paragraphs are very long e.g., the first paragraph in the Introduction is a whole page. The paper would be a lot clearer and easier to read if each paragraph only dealt with one topic and was thus shorter.

We divided the first paragraph into two paragraphs.

2) The Introduction could be improved:

- It should be clearer what's new in this study compared to previous estimates. That message is indirectly in the introduction but should be made more explicitly (see also my major comment).

We rephrased the last paragraph of the introduction to directly state what's new here:

“To date, the C_{ant} estimates are mainly based on methods that rely on scarce but valuable ship-based measurements of carbonate system parameters (carbon-based methods e.g., Gruber et al., 2019; Müller et al., 2023; Pérez et al., 2013; Sabine et al., 2004; Woosley et al., 2016) or transient tracers such as CFCs (transient tracer-based methods, e.g., Raimondi et al., 2021; Waugh et al., 2006). However, studying the spatio-temporal evolution of oceanic C_{ant} storage and understanding the processes involved is a crucial challenge that requires a more detailed view of the upper and deep C_{ant} distribution and of its main pathways into the ocean interior. Additionally, the distribution of C_{ant} on timescales shorter than GO-SHIP cruises is necessary to understand the effect of CO_2 emissions reduction and carbon removal strategies on the ocean. Considering the unrivaled spatio-temporal sampling provided by the Argo- O_2 network (Roemmich et al., 2019), the purpose of this study is to demonstrate that Argo- O_2 data combined with existing neural networks (i.e., ESPER_NN (Carter et al., 2021), CANYON-B and CONTENT (Bittig, Steinhoff, et al., 2018)) and a back-calculation method can be used to obtain reliable C_{ant} estimate at the finest spatio-temporal scale to date.”

- Reading the introduction, I assumed that the authors had developed a new neural network approach and wondered why they didn't use existing frameworks such as ESPER, CONTENT, and CANYON. In the Methods section, it becomes clear, that they use those existing frameworks. This should be mentioned in the introduction, even if it is just by adding the words 'previously developed' to neural networks (this was done in the abstract with 'existing neural networks', but I missed it in the introduction). Consider explicitly mentioning CANYON, CONTENT, and ESPER in the introduction to make it even clearer.

Indeed we used existing neural networks. We modified the sentence:

“Considering the unrivaled spatio-temporal sampling provided by the Argo- O_2 network (Roemmich et al., 2019), the purpose of this study is to demonstrate that Argo- O_2 data combined with existing neural networks (i.e., ESPER_NN (Carter et al., 2021), CANYON-B and CONTENT (Bittig, Steinhoff, et al., 2018)) and a back-calculation method can be used to obtain reliable C_{ant} estimate at the finest spatio-temporal scale to date.”

3) I find the Discussion a bit confusing, and it doesn't flow well because it mixes insights from previous studies with insights from this study, and the separation is not always clear. This section could be more powerful if it was more clearly separated (as well as shorter paragraphs, as mentioned above).

We revised the Discussion and added a couple of sentences on the limits of neural networks. The Discussion is now clearly divided into four paragraphs: The 1st paragraph summarizes our findings.

The 2nd paragraph compares our findings with previous studies.

The 3rd paragraph acknowledges how our methodology can be improved. Specifically, we state that we can only guarantee the validity of our approach in the case-study region and for the period (1970-2020) over which the neural networks have been trained.

The 4th paragraph concludes on how C_{ant} can be better represented in Earth System models.

Specific Comments

L.24: But this study also doesn't address the deep open beyond 2000 m. Consider rephrasing to 'sub-surface' or 'upper ocean' instead of 'deep'

We rephrased by "sub-surface".

L.24: We do know a bit about spatio-temporal variability of C_{ant} in the North Atlantic from Sabine et al., 2004, Gruber et al., 2019, and Mueller et al. 2023. It's just that those studies are on decadal timescales, while this study is looking at shorter timescales.

We rephrased by "their spatio-temporal variability on short timescales (<1 year, <10 km) remain largely unresolved."

L35: Can C_{ant} have an amplitude? Consider rephrasing, e.g., to "amount of C_{ant} " if that's what you mean.

We rephrased by "the amount of C_{ant} ".

L52: Remove 'total' (as the number provided is per year)

We removed "total".

L94: Consider adding also the importance of understanding C_{ant} in light of (hopefully) CO₂ emission reductions and carbon removal strategies.

We added the sentence "Additionally, the distribution of C_{ant} on timescales shorter than GO-SHIP cruises is necessary to understand the effect of CO₂ emissions reduction and carbon removal strategies on the ocean."

L97: Not only the deep, also the upper C_{ant} . Rephrase e.g., to "upper ocean"

We rephrased by "of the upper and deep C_{ant} distribution".

L100: Add a reference to this new study: Mueller et al., 2023

We added the reference of Müller et al. (2023).

L110: Add that 'seasonally unbiased' is compared to ship data

We added "compared to ship-based measurements" at the end of the sentence.

Fig.1: Add that the purple arrows are between the upper and deep

In the caption of Fig. 1 we added the sentence:

"The purple arrow represents ocean circulation between the upper and lower limb of the AMOC."
"

L242: It should be added if the visual inspection ever disagrees with the MLD estimate. And if so, what was done when the visual inspection disagreed with the MLD estimate?

We added the sentence:

"When the MLD values disagreed between the "threshold method" and the visual inspection, we kept the value determined by visual inspection."

L274: Why was not a more up-to-date method used, e.g., AVISO META by Pegliasco et al. (2022). (It's not a big deal and doesn't need to be changed, I'm just curious).

We started our analysis with the database of Faghmous et al. (2015) and the improvements introduced by Pegliasco et al. (2022) only concern the coastal regions. Since our study is based on Argo-O₂ floats traveling in the open ocean, and not coastal areas, we kept the method of Faghmous et al. (2015).

L296: floats' (not float's)

Changed

I hope you find this review helpful. Best wishes and all the best with the publication of this important and exciting manuscript.

We thank the reviewer again for their valuable comments that improve the quality of the manuscript.

References

Broullón, D., Pérez, F. F., Velo, A., Hoppema, M., Olsen, A., Takahashi, T., Key, R. M., Tanhua, T., González-Dávila, M., Jeansson, E., Kozyr, A., & van Heuven, S. M. A. C. (2019). A global monthly climatology of total alkalinity: A neural network approach. *Earth System Science Data*, 11(3), 1109–1127. <https://doi.org/10.5194/essd-11-1109-2019>

Broullón, D., Pérez, F. F., Velo, A., Hoppema, M., Olsen, A., Takahashi, T., Key, R. M., Tanhua, T., Santana-Casiano, J. M., & Kozyr, A. (2020). A global monthly climatology of oceanic total dissolved inorganic carbon: A neural network approach. *Earth System Science Data*, 12(3), 1725–1743. <https://doi.org/10.5194/essd-12-1725-2020>

Gruber, N., Landschützer, P., & Lovenduski, N. S. (2019). The Variable Southern Ocean Carbon Sink. *Annu. Rev. Mar. Sci.* 2019.

Keppler, L., Landschützer, P., Gruber, N., Lauvset, S. K., & Stemmler, I. (2020). Seasonal Carbon Dynamics in the Near-Global Ocean. *Global Biogeochemical Cycles*, 34(12), e2020GB006571. <https://doi.org/10.1029/2020GB006571>

Keppler, L., Landschützer, P., Lauvset, S. K., & Gruber, N. (2023). Recent Trends and Variability in the Oceanic Storage of Dissolved Inorganic Carbon. *Global Biogeochemical Cycles*, 37(5), e2022GB007677. <https://doi.org/10.1029/2022GB007677>

Müller, J. D., Gruber, N., Carter, B., Feely, R., Ishii, M., Lange, N., Lauvset, S. K., Murata, A., Olsen, A., Pérez, F. F., Sabine, C., Tanhua, T., Wanninkhof, R., & Zhu, D. (2023). Decadal Trends in the Oceanic Storage of Anthropogenic Carbon From 1994 to 2014. *AGU Advances*, 4(4), e2023AV000875. <https://doi.org/10.1029/2023AV000875>

Pegliasco, C., Delepouille, A., Mason, E., Morrow, R., Faugère, Y., & Dibarboure, G. (2022). META3.1exp: A new global mesoscale eddy trajectory atlas derived from altimetry. *Earth System Science Data*, 14(3), 1087–1107. <https://doi.org/10.5194/essd-14-1087-2022>

Roemmich, D., & Gilson, J. (2009). The 2004-2008 mean and annual cycle of temperature, salinity, and steric height in the global ocean from the Argo Program. *Progress in Oceanography*, 82(2), 81–100. <https://doi.org/10.1016/j.pocean.2009.03.004>

Sabine, C. L., Feely, R. A., Gruber, N., Key, R. M., Lee, K., Bullister, J. L., Wanninkhof, R., Wong, C. S., Wallace, D. W. R., Tilbrook, B., Millero, F. J., Peng, T. H., Kozyr, A., Ono, T., & Rios, A. F. (2004). The oceanic sink for anthropogenic CO₂. *Science*, 305(5682), 367–371. <https://doi.org/10.1126/science.1097403>

Sharp, J. D., Fassbender, A. J., Carter, B. R., Johnson, G. C., Schultz, C., & Dunne, J. P. (2023). GOBAI-O2: Temporally and spatially resolved fields of ocean interior dissolved oxygen over nearly two decades. *Earth System Science Data Discussions*, 1–46. <https://doi.org/10.5194/essd-2022-308>

Biló, T. C., Straneo, F., Holte, J., & Le Bras, I. A. (2022). Arrival of New great salinity anomaly weakens convection in the Irminger Sea. *Geophysical Research Letters*, 49(11), e2022GL098857.

Bittig, H. C., Steinhoff, T., Claustre, H., Fiedler, B., Williams, N. L., Sauzède, R., ... & Gattuso, J. P. (2018). An alternative to static climatologies: Robust estimation of open ocean CO₂ variables and nutrient concentrations from T, S, and O₂ data using Bayesian neural networks. *Frontiers in Marine Science*, 5, 328.

Broullón, D., Pérez, F. F., Velo, A., Hoppema, M., Olsen, A., Takahashi, T., ... & van Heuven, S. M. (2019). A global monthly climatology of total alkalinity: a neural network approach. *Earth System Science Data*, 11(3), 1109-1127.

Keppler, L., Landschützer, P., Lauvset, S. K., & Gruber, N. (2023). MOBO-DIC: recent trends and variability in the oceanic storage of dissolved inorganic carbon.

Lozier, M. S., McCartney, M. S., & Owens, W. B. (1994). Anomalous anomalies in averaged hydrographic data. *Journal of Physical Oceanography*, 24(12), 2624-2638.

Metropolis, N., & Ulam, S. (1949). The monte carlo method. *Journal of the American statistical association*, 44(247), 335-341.

Müller, J. D., Gruber, N., Carter, B., Feely, R., Ishii, M., Lange, N., et al. (2023). Decadal trends in the oceanic storage of anthropogenic carbon from 1994 to 2014. *AGU Advances*, 4, e2023AV000875.

Olsen, A., Lange, N., Key, R. M., Tanhua, T., Bittig, H. C., Kozyr, A., ... & Woosley, R. J. (2020). An updated version of the global interior ocean biogeochemical data product, GLODAPv2. 2020. *Earth System Science Data*, 12(4), 3653-3678.

Roemmich, D., & Gilson, J. (2009). The 2004–2008 mean and annual cycle of temperature, salinity, and steric height in the global ocean from the Argo Program. *Progress in oceanography*, 52(2), 81-100.

Shi, J., Wang, J., Ren, Z., Tang, C., & Huang, F. (2023). Cold blobs in the subpolar North Atlantic: seasonality, spatial pattern, and driving mechanisms. *Ocean Dynamics*, 1-12.

Wong, A. P., Wijffels, S. E., Riser, S. C., Pouliquen, S., Hosoda, S., Roemmich, D., ... & Park, H. M. (2020). Argo data 1999–2019: Two million temperature-salinity profiles and subsurface velocity observations from a global array of profiling floats. *Frontiers in Marine Science*, 7, 700.

REVIEWERS' COMMENTS

Reviewer #5 (Remarks to the Author):

General Comments:

Asselot and coauthors use Argo floats in the North Atlantic that were equipped with dissolved oxygen sensors, along with previously developed seawater property estimation neural networks and back-calculation methods, to estimate anthropogenic carbon (Cant) along the float paths. They describe the evolution of the Cant signal in North Atlantic water masses. In particular, the authors show a stepwise deepening and dilution of Cant during the formation of Subpolar Mode Water and Labrador Sea Water. They identify the Reykjanes Ridge as a critical bathymetric feature separating the Iceland Basin, where Cant mixes to about 600 dbar, from the Irminger Basin, where Cant mixes to about 1400 dbar. They also speculate about the role of mesoscale eddies in driving periodic downward pulses of Cant in the region. In general, the authors make the point that applying their method to Argo-O₂ floats represents a promising pathway for retrieving Cant at high spatial and temporal resolution.

One note is that the authors are incorrect in their impression of the Roemmich and Gilson (2009), Keppler et al. (2023), and Sharp et al. (2023) data products of temperature/salinity, DIC, and O₂, respectively. Each of those products do indeed offer gridded snapshots for each month over the periods mentioned, not just an average climatology over those periods. So the relevant overlapping time periods could be extracted from each product. Still, I recognize the value of the quasi-lagrangian floats for tracking water mass movements and their Cant accumulation. Also, the products will smooth over fine-scale features to some degree compared to direct float measurements, and the temporal resolution of the float profiles is three times greater than the monthly products. These considerations could be added as additional justifications for using direct float measurements rather than monthly data products, somewhere around line 165.

Overall, the authors' response to reviewer comments appears sufficient. Though Reviewer #4 raises concerns about the uncertainty of the method, the authors do indeed interpret their results in the context of their assessed uncertainty (e.g., lines 233, 417, 568). In addition, Reviewer #4 raises a hypothetical about using the authors' method to evaluate interannual changes in Cant. However, this is not attempted by the authors in the present study, as they merely evaluate Cant patterns in the North Atlantic and interpret them in relation to associated data and what is known about water mass formation in the region.

Specific Comments:

Make sure the order of Supplementary Figures corresponds to the order in which they are introduced in the manuscript.

155: Why was ESPER_NN chosen over ESPER_LIR or ESPER_Mixed? This should be briefly acknowledged and explained.

158-159: Supplementary Table 1 is referenced here but it's not immediately obvious from the table how the uncertainties in derived parameters were calculated.

160: It should be made clear why ESPER was used for nutrients whereas CANYON-B and CONTENT were used for carbonate system variables.

190-193: Make sure to note here what other methods the phi_CTO method is being compared to.

547-549: This study doesn't necessarily prove the point stated in these lines, as retrieving 3D estimates of those variables from float data has been done many times before. The novel aspect of the study is more so in using those derived variables for Cant analysis.

Supp. Table 1: More information should be given in the caption about how these values were obtained.

REVIEWERS' COMMENTS

Reviewer #5 (Remarks to the Author):

General Comments:

Asselot and coauthors use Argo floats in the North Atlantic that were equipped with dissolved oxygen sensors, along with previously developed seawater property estimation neural networks and back-calculation methods, to estimate anthropogenic carbon (Cant) along the float paths. They describe the evolution of the Cant signal in North Atlantic water masses. In particular, the authors show a stepwise deepening and dilution of Cant during the formation of Subpolar Mode Water and Labrador Sea Water. They identify the Reykjanes Ridge as a critical bathymetric feature separating the Iceland Basin, where Cant mixes to about 600 dbar, from the Irminger Basin, where Cant mixes to about 1400 dbar. They also speculate about the role of mesoscale eddies in driving periodic downward pulses of Cant in the region. In general, the authors make the point that applying their method to Argo-O₂ floats represents a promising pathway for retrieving Cant at high spatial and temporal resolution.

We would like to thank the reviewer for their valuable comments that improved the quality of the manuscript.

One note is that the authors are incorrect in their impression of the Roemmich and Gilson (2009), Keppler et al. (2023), and Sharp et al. (2023) data products of temperature/salinity, DIC, and O₂, respectively. Each of those products do indeed offer gridded snapshots for each month over the periods mentioned, not just an average climatology over those periods. So the relevant overlapping time periods could be extracted from each product. Still, I recognize the value of the quasi-lagrangian floats for tracking water mass movements and their Cant accumulation. Also, the products will smooth over fine-scale features to some degree compared to direct float measurements, and the temporal resolution of the float profiles is three times greater than the monthly products. These considerations could be added as additional justifications for using direct float measurements rather than monthly data products, somewhere around line 165.

At the end of the “Estimating biogeochemical variables with neural networks” section we added the sentences:

“In contrast, neural networks, such as ESPER_NN and CANYON-B, derive biogeochemical variables based on water mass characteristics, hence they can cope with such changes. In particular, the use of Argo-O₂ float data as input to the neural networks reinforces this water-mass change tracking capability, due to the quasi-lagrangian behavior of the Argo-O₂ floats and their temporal resolution, three times greater than monthly climatological products.”

Overall, the authors' response to reviewer comments appears sufficient. Though Reviewer #4 raises concerns about the uncertainty of the method, the authors do indeed interpret their results in the context of their assessed uncertainty (e.g., lines 233, 417, 568). In addition, Reviewer #4 raises a hypothetical about using the authors' method to evaluate interannual changes in Cant. However, this is not attempted by the authors in the present study, as they merely evaluate Cant patterns in the North Atlantic and interpret them in relation to associated data and what is known about water mass formation in the region.

We thank the reviewer for this comment.

Specific Comments:

Make sure the order of Supplementary Figures corresponds to the order in which they are introduced in the manuscript.

We modified the order of the Supplementary Figures.

155: Why was ESPER_NN chosen over ESPER_LIR or ESPER_Mixed? This should be briefly acknowledged and explained.

We added the sentences:

“ESPER_NN was adopted over other ESPER methods (i.e. ESPER_LIR and ESPER_Mixed), because it gives the lowest biases and root mean square errors over the global ocean for the predicted macronutrients (Carter et al., 2021).”

158-159: Supplementary Table 1 is referenced here but it's not immediately obvious from the table how the uncertainties in derived parameters were calculated.

We changed the caption of Supplementary Table 1:

“Uncertainties on the input variables to calculate Cant for the 3 Argo floats. The uncertainties of pressure, potential temperature, salinity and oxygen after delayed-mode correction are provided by the Argo program. The uncertainties of silicate, nitrate and phosphate correspond to the mean ESPER_NN uncertainties (Carter et al., 2021). The uncertainties of alkalinity and dissolved inorganic carbon represent the mean CONTENT uncertainties (Bittig, Steinhoff, et al., 2018). See text above for more explanations. ”

The text above Supplementary Table 1 shortly indicates how the ESPER_NN and CONTENT uncertainties are computed. We refer the reader to the original neural networks' papers for more information.

160: It should be made clear why ESPER was used for nutrients whereas CANYON-B and CONTENT were used for carbonate system variables.

In the section called “Estimating biogeochemical variables with neural networks”, we completed the text as follow:

“However, in the North Atlantic Ocean, ESPER_NN gives uncertainties of ~1.3% for the predicted carbonate variables (A_T and DIC), which is higher than previous NN. Consequently, A_T and DIC were computed with CANYON-B (Bittig, Steinhoff, et al., 2018) and the outputs of CANYON-B were passed through the CONTENT routine. This routine ensures consistency between carbonate variables and thus reduces the uncertainties of the carbonate system variables to ~0.5% for AT and DIC (Supplementary Table 1). ”

190-193: Make sure to note here what other methods the phi_CTO method is being compared to.

We rephrased the sentence as:

“A study comparing observational methods to estimate C_{ant} in the Atlantic Ocean, including the TTD (Vaughan et al., 2006), the TrOCA (Touratier et al., 2007), the C_{IPSL}^o (Lo Monaco et al., 2005), the ΔC^* (Gruber et al., 1996) and the ϕC_T^o method (Pérez et al., 2008), proved that the latter provided the closest value to the average of all methods for the whole latitudinal range (Vazquez-Rodriguez et al., 2009).”

547-549: This study doesn't necessarily prove the point stated in these lines, as retrieving 3D estimates of those variables from float data has been done many times before. The novel aspect of the study is more so in using those derived variables for C_{ant} analysis.

We rephrased these lines as:

“Our case-study has proved that neural networks combined with high-quality in situ Argo- O_2 measurements and a back-calculation method can effectively be used to retrieve C_{ant} concentration through the three-dimensional estimates of the oceanic variables (nutrients, DIC, total alkalinity).”

Supp. Table 1: More information should be given in the caption about how these values were obtained.

See our answer to comment on line 158-159